# Non-CG methylation and multiple histone profiles associate child abuse with immune and small GTPase dysregulation

Pierre-Eric Lutz [1,8,11], Marc-Aurèle Chay[1,11], Alain Pacis[2], Gary G. Chen[1], Zahia Aouabed[1], Elisabetta Maffioletti[3], Jean-François Théroux[1], Jean-Christophe Grenier [2,9], Jennie Yang[1], Maria Aguirre[4], Carl Ernst [1,5], Adriana Redensek[6], Léon C. van Kempen[4], Ipek Yalcin[7], Tony Kwan [6], Naguib Mechawar[1,5], Tomi Pastinen[6,10] & Gustavo Turecki [1,5✉]

Early-life adversity (ELA) is a major predictor of psychopathology, and is thought to increase lifetime risk by epigenetically regulating the genome. Here, focusing on the lateral amygdala, a major brain site for emotional homeostasis, we describe molecular cross-talk among multiple mechanisms of genomic regulation, including 6 histone marks and DNA methylation, and the transcriptome, in subjects with a history of ELA and controls. In the healthy brain tissue, we first uncover interactions between different histone marks and non-CG methylation in the CAC context. Additionally, we find that ELA associates with methylomic changes that are as frequent in the CAC as in the canonical CG context, while these two forms of plasticity occur in sharply distinct genomic regions, features, and chromatin states. Combining these multiple data indicates that immune-related and small GTPase signaling pathways are most consistently impaired in the amygdala of ELA individuals. Overall, this work provides insights into genomic brain regulation as a function of early-life experience.

[1] McGill Group for Suicide Studies, Douglas Mental Health University Institute, McGill University, Montréal, Canada. [2] Department of Genetics, CHU Sainte-Justine Research Center, Montréal, Canada. [3] Genetics Unit, IRCCS Istituto Centro San Giovanni di Dio Fatebenefratelli, Brescia, Italy. [4] Segal Cancer Centre, Lady Davis Institute, Jewish General Hospital, McGill University, Montréal, Canada. [5] Department of Psychiatry, McGill University, Montréal, Canada. [6] Department of Human Genetics, McGill University, Montréal, Canada. [7] Centre National de la Recherche Scientifique, Institut des Neurosciences Cellulaires et Intégratives, Université de Strasbourg, Fédération de Médecine Translationnelle de Strasbourg, Strasbourg, France. [8] Present address: Centre National de la Recherche Scientifique, Institut des Neurosciences Cellulaires et Intégratives, Université de Strasbourg, Fédération de Médecine Translationnelle de Strasbourg, Strasbourg, France. [9] Present address: Institut de Cardiologie de Montréal, Montréal, Canada. [10] Present address: Center for Pediatric Genomic Medicine, University of Missouri-Kansas City School of Medicine, Kansas City, MO, USA. [11] These authors contributed equally: Pierre-Eric Lutz, Marc-Aurèle Chay. ✉email: gustavo.turecki@mcgill.ca

Early-life adversity (ELA), including sexual and physical abuse, as well as other forms of child maltreatment, is a major public health problem that affects children of all socio-economic backgrounds[1]. ELA is a strong predictor of increased lifetime risk of negative mental health outcomes, including depressive disorders[2]. Among other findings, a large number of studies suggest an association between ELA and morphological and functional changes in the amygdala[3], a brain structure critically involved in emotional regulation[4]. It is possible, thus, that amygdala changes observed in individuals who experienced ELA may contribute to increased risk of psychopathology.

The amygdala is composed of interconnected nuclei, among which the basal and lateral sub-divisions are responsible for receiving and integrating external information. In turn, these nuclei innervate the central amygdala, the primary nucleus projecting outside the amygdalar complex to mediate behavioral outputs[4]. While specific properties of these nuclei remain difficult to assess in humans, animal studies indicate that the basal and lateral sub-divisions exhibit differential responsivity to stress, in particular as a function of the developmental timing of exposure (adolescence versus adulthood)[5,6]. Here, we focused on homogeneous, carefully dissected tissue from the human lateral amygdala.

Childhood is a sensitive period during which the brain is more responsive to the effect of life experiences[7]. Proper emotional development is contingent on the availability of a supportive caregiver, with whom children develop secure attachments[8]. On the other hand, ELA signals an unreliable environment that triggers adaptive responses and deprives the organism of essential experience. A growing body of evidence now supports the hypothesis that epigenetic mechanisms play a major role in the persistent impact of ELA on gene expression and behavior[9]. While DNA methylation has received considerable attention, available data also point toward histone modifications as another critical and possibly interacting factor[9].

Therefore, in this study, we conduct a comprehensive characterization of epigenetic changes occurring in individuals with a history of severe ELA and carry out genome-wide investigations of multiple epigenetic layers, and their cross-talk. Using post-mortem brain tissue from a well-defined cohort of depressed individuals with histories of ELA, and controls with no such history, we characterize six histone marks, DNA methylation, as well as their final endpoint at the gene expression level. We first generate data for six histone modifications: H3K4me1, H3K4me3, H3K27ac, H3K36me3, H3K9me3, and H3K27me3[10], using chromatin immunoprecipitation sequencing (ChIP-Seq). This allows us to create high-resolution maps for each mark, and to define chromatin states throughout the epigenome. In parallel, we characterize DNA methylation using whole-genome bisulfite sequencing (WGBS). While previous studies in psychiatry focused on the canonical form of DNA methylation that occurs at CG dinucleotides (mCG), here we investigate both CG and non-CG contexts. Indeed, recent data has shown that non-CG methylation is not restricted to stem cells, and can be detected in brain tissue at even higher levels[11]. Available evidence also indicates that it progressively accumulates, preferentially in neurons, during the first decade of life[12,13], a period when ELA typically occurs. Thus, we postulate that changes in non-CG methylation might contribute to lifelong consequences of ELA, and focus in particular on the CAC context, where non-CG methylation is most abundant. Our results indicate that ELA leaves distinct, albeit equally frequent, traces at CG and CAC sites. Further, analyses of all epigenetic layers and the transcriptome converge to identify immune system processes and small GTPases as critical pathways associated with ELA.

Altogether, these data uncover previously unforeseen sources of epigenetic and transcriptomic plasticity, which may contribute to the severe and lifelong impact of ELA on behavioral regulation, and the risk of depression.

## Results

**Histone landscapes.** Six histone modifications were assessed in depressed subjects with histories of ELA, and healthy controls (C) with no such history (Supplementary Tables 1 and 2). Because of the small size of the lateral amygdala, and the significant amount of tissue required for multiple immuno-precipitations and ChIP-seq analysis of six marks, tissues were distributed into 7 ELA and 4 C pools (see Supplementary Table 3). In contrast, WGBS and RNA-Seq data (see below) were generated for each individual sample (C, $n = 17$; ELA, $n = 21$). Following the International Human Epigenome Consortium (IHEC) procedures, we achieved >60 and >30 million reads for broad (H3K4me1, H3K36me3, H3K27me3, and H3K9me3) and narrow (H3K27ac and H3K4me3) histone marks, respectively (4.0 billion reads total; Supplementary Fig. 1a and Supplementary Data 1). Quality controls confirmed that all samples for the two narrow marks showed relative and normalized strand cross-correlations greater than 0.8 and 1.05 (Supplementary Fig. 1b), respectively, according to expectations[14]. Relative to genes (Fig. 1a, c), reads obtained for H3K27ac, H3K4me3 and H3K4me1 were strongly enriched around Transcription Start Sites (TSS), while H3K27me3 and H3K36me3 showed antagonistic distributions, consistent with patterns seen in other tissues[10]. Samples clustered by histone mark, with a strong distinction between activating and repressive marks (Fig. 1b). To investigate the tissue specificity of our dataset, we compared it with data from other brain regions and blood tissue (Supplementary Fig. 2). For each modification, we observed higher correlations among amygdalar samples ($r = 0.75-0.92$ across the six marks) than when compared with samples from other brain regions ($r = 0.51-0.81$), and even lower correlations with blood mononuclear cells ($r = 0.35-0.64$), consistent with the role of histones in tissue identity.

We next investigated relationships between histones and gene expression (Fig. 1d). As expected, we observed activating functions for H3K27ac, H3K4me1, H3K36me3, and H3K4me3, and repressive functions for H3K27me3 and H3K9me3. Distinct correlative profiles were found between marks along the spectrum of gene expression, indicating that multiple marks likely better predict gene expression than individual ones. Comparisons between ELA and C groups found no significant overall differences in terms of read distribution (Fig. 1c) or relationship to gene expression (Fig. 1d), indicating that ELA does not globally reconfigure amygdalar histone landscapes.

Considering that different combinations of histone modifications define so-called "chromatin states", we then conducted an integrative analysis of all marks using ChromHMM[15]. Maps of chromatin states were generated as described previously[16], with each state corresponding to a distinct combination of individual marks. This unbiased approach defined a consensus map (corresponding to regions showing ≥50% agreement across all samples; see Fig. 1e, Supplementary Fig. 3a–d and "Methods") consistent with studies in the brain and other tissues: for example[16–18], regions defined by H3K27ac and H3K4me1, or by H3K36me3, corresponded to known enhancers (Gen Enh and Enh) and transcribed regions (Str-Trans and Wk-Trans), respectively (Supplementary Fig. 3e, f)[19]. Compared with known genomic features (Fig. 1f), this map showed expected enrichments of promoter chromatin states (Act, Wk, or Flk-Prom) at transcription start sites and CpG islands, and of transcription states (Str-Trans and Wk-Trans) within genes. Finally, the

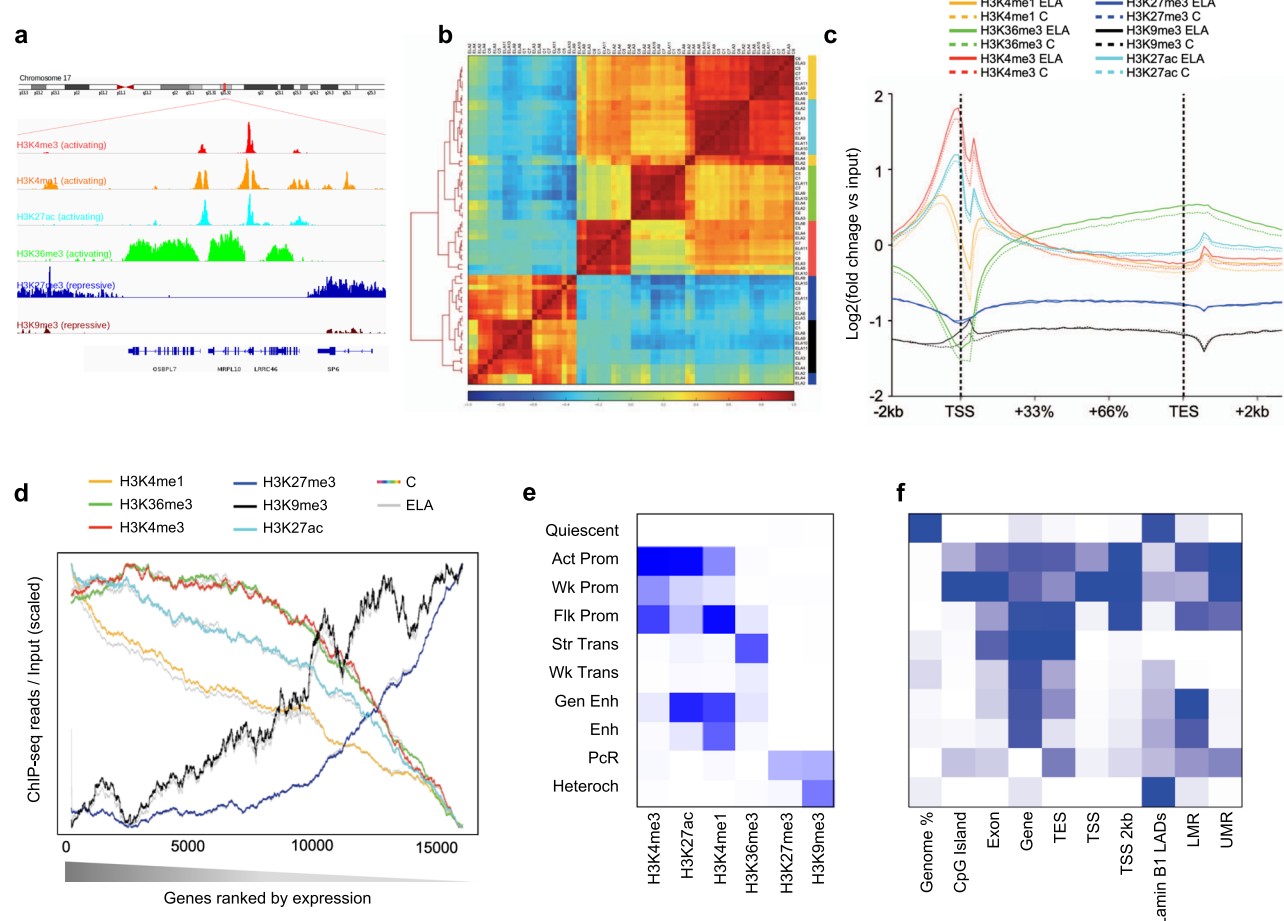

**Fig. 1 Characterization of six histone post-translational modifications in the human brain lateral amygdala. a** Snapshot of typical ChIP-seq read distribution for the six histone marks. **b** Unsupervised hierarchical clustering using Pearson correlations for all marks. Correlations were computed using read number per 10 kb-bins across the whole genome and normalized to input and library size. Note the expected separation between activating (H3K27ac, H3K36me3, H3K4me1, H3K4me3) and repressive (H3K27me3, H3K9me3) marks. **c** Average enrichment over the input of ChIP-seq reads across all gene bodies and their flanking regions ($+/-$ 2 kilobases, kb) in the human genome, for each histone mark. Note the expected biphasic distribution of reads around the TSS for H3K27ac, H3K4me3, and H3K4me1. No significant differences were observed for any mark across C and ELA groups (two-sided two-way repeated-measures ANOVA, group effects: H3K4me1, $P = 0.89$; H3K36me3, $P = 0.87$; H3K4me3, $P = 0.64$; H3K27me3, $P = 0.35$; H3K9me3, $P = 0.88$; H3K27ac, $P = 0.86$). Averages for the healthy controls group (C) are shown as dashed lines, while averages for the early-life adversity group (ELA) are shown as solid lines. **d** Average enrichment of reads over gene bodies (for H3K27me3, H3K36me3, H3K4me1, and H3K9me3) or TSS $+/-$ 1 kb (for H3K27ac and H3K4me3) for all genes ranked from most highly (left) to least (right) expressed. Strongly significant effects of gene ranking on ChIP-Seq reads were observed for all marks ($P < 0.0001$). Again, no difference was observed as a function of ELA for any group (two-sided two-way repeated-measures ANOVA, group effects: H3K4me1, $P = 0.66$; H3K36me3, $P = 0.67$; H3K4me3, $P = 0.98$; H3K27me3, $P = 0.31$; H3K9me3, $P = 0.74$; H3K27ac, $P = 0.48$). **e** ChromHMM emission parameters (see main text and "Methods") for the 10-state model of chromatin generated using data from the six histone marks, at a resolution of 200 bp, as described previously[16]. Maps of chromatin states have already been characterized in other brain regions (e.g., cingulate cortex, caudate nucleus, substantia nigra[49]) but, to our knowledge, not in the amygdala. **f** Intersections of chromatin states with gene features (from RefSeq) and methylomic features (lowly methylated and unmethylated regions, LMR and UMR, defined using methylseekR; see "Methods") were computed using chromHMM's OverlapEnrichment function. As expected, CpG-dense UMRs mostly overlapped with Promoter chromatin states, while LMRs associated with more diverse chromatin states, including Enhancers (Fig. 1f and Supplementary Fig. 10a), consistent with their role as distant regulatory sites[31]. Act-Prom active promoter, Enh enhancer, Flk-Prom flanking promoter, Heterochr heterochromatin, LADs lamina-associated domains, PcR polycomb repressed, Str-Enh strong enhancer, Str-Trans strong transcription, TES transcription end site, TSS transcription start site, Wk-Prom weak promoter, Wk-Trans weak transcription. Source data are provided as a Source Data file.

chromatin states exhibit expected correlations with gene expression (Supplementary Fig. 4). As detailed below, these maps allowed us to characterize cross-talks between chromatin and DNA methylation, and differences between groups.

**CG and non-CG methylation patterns**. We used WGBS to characterize the amygdala methylome (C, $n = 17$; ELA, $n = 21$). Rates of bisulfite conversion, sequencing depth, and library diversity met IHEC standards and were similar across groups (Supplementary Fig. 5a–d). In this large dataset, >13 million CGs showed an average coverage ≥5 in the cohort (Supplementary Fig. 5e), which favorably compares with recent human brain studies in terms of sample size[20] or CGs covered[21,22].

Because non-CG methylation is enriched in mammalian brains[11,23], we first computed average genome-wide levels of methylation in multiple cytosine contexts. Focusing on three-letter contexts (Fig. 2a), we observed that, as expected,

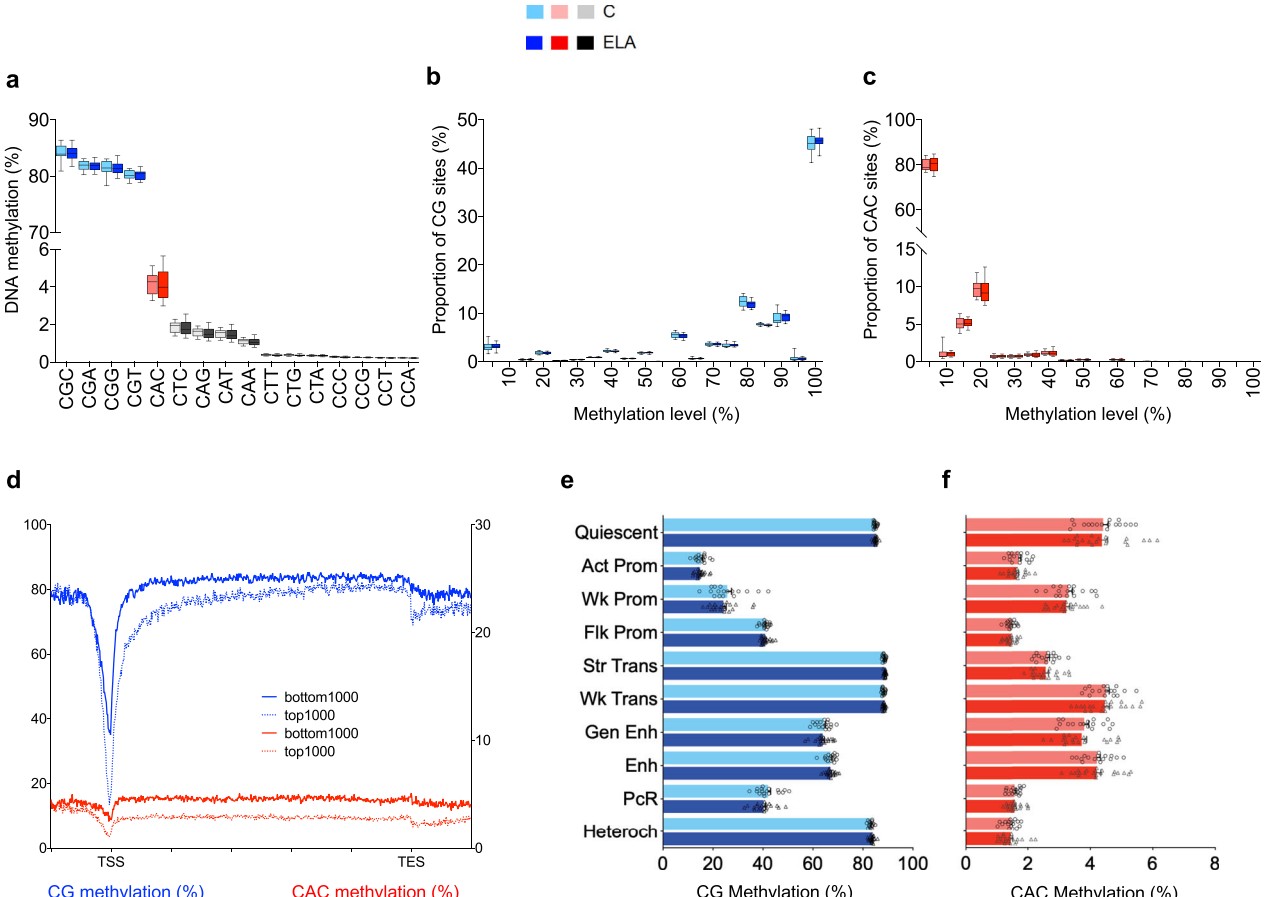

**Fig. 2 Characterization of non-CG methylation in the human brain lateral amygdala. a** Average genome-wide levels of DNA methylation were measured among the sixteen three-letter cytosine contexts (CNN, where N stands for any base) in the human brain lateral amygdala, using whole-genome bisulfite sequencing. While highest DNA methylation levels were observed in the four CGN contexts (CGC: 84.1 ± 0.2%; CGA: 81.9 ± 0.1%; CGC: 81.4 ± 0.2%, CGT: 80.2 ± 0.1%; mean ± sem in the whole cohort), detectable non-CG methylation was also observed in CHN context (where H stands for A, C, and T), most notably at CAC sites (4.1 ± 0.1% in combined control, C, and early-life adversity, ELA, groups; $n = 38$ subjects total), with no detectable differences between groups for any context (two-way repeated-measures ANOVA; group effect: [$F(1,36) = 0.12$; $P = 0.73$]). **b** DNA methylation in the CG context mostly corresponded to highly methylated sites. In contrast, as previously described in the mouse hippocampus[29], most CAC sites were unmethylated (**c**), with only a minority of them showing low methylation levels, between 10 and 20% ($n = 38$ subjects). This likely reflects the fact that non-CG methylation does not occur in all cell types, and is notably enriched in neuronal cells and, to a lesser extent, in glial cells[11]. In the CG or CAC contexts (two-way ANOVA; group effect: CG, [$F(1,720) = 5.0E-11$; $P > 0.99$]; CAC, [$F(1,36) = 0$; $P > 0.99$]), ELA did not associate with any significant change in these global distributions. Box plots show median and interquartile range, with whiskers representing minimum and maximum values. **d** In both contexts, patterns of DNA methylation along gene bodies showed the expected anti-correlation with gene expression, as shown here comparing 1000 most highly (top 1000) or lowly (bottom 1000) expressed genes, consistent with previous rodent data. In the CG (**e**) or CAC (**f**) contexts, no difference in DNA methylation levels was observed between C and ELA groups for any chromatin state (values are mean ± sem in each C or ELA group; $n = 17$ and 21 subjects, respectively). We observed, however, dissociations in the relationship of DNA methylation and histone marks across the CG and CAC contexts (see main text). Values are mean ± sem. Act-Prom active promoter, Enh enhancer, Flk-Prom flanking promoter, Heterochr heterochromatin, PcR polycomb repressed, Str-Enh strong enhancer, Str-Trans strong transcription, TES transcription end site, TSS transcription start site, Wk-Prom weak promoter, Wk-Trans weak transcription. Source data are provided as a Source Data file.

methylation levels were highly variable among the 16 possibilities (two-way ANOVA; context effect: [$F(15,540) = 196283$; $P < 0.0001$]), with much higher methylation levels in the CGN contexts than in the 12 non-CG contexts. Of note, no difference was found in overall methylation between groups ([$F(1,36) = 0.12$; $P = 0.73$]), indicating that ELA does not associate with a global dysregulation of the methylome. Among non-CG contexts, as previously described by others in mice[24] or humans[25,26] (Supplementary Fig. 6a, b), methylation levels were highest at CACs (4.1 ± 0.1%), followed by a group of contexts between 1.8 and 1.1% (CTC, CAG, CAT, and CAA), and remaining ones below 0.4%. Considering that methylation at CA[27] or CAC[28] sites may have specific functions in the brain, and because CAC

methylation (hereafter mCAC) was most abundant, we focused on this context.

We first compared mCG and mCAC. While CG sites were highly methylated, CAC (Fig. 2b, c) or other non-CG (Supplementary Fig. 6c) sites were mostly unmethylated, with a minority of them showing methylation levels between 10 and 20%, consistent with mouse data[29]. Regarding distinct genomic features and chromosomal location, we confirmed that (i) while mCG is lower within promoters, this effect is much less pronounced for mCAC (Supplementary Fig. 7a)[11]; (ii) compared with CGs[30], depletion of methylation from pericentromeric regions is even stronger at CACs, and (iii) as expected, methylation levels were very low in both contexts in the

mitochondrial genome (Supplementary Fig. 7b). We also confronted methylation data with gene expression, regardless of group status, and found the expected anti-correlation in both contexts (Fig. 2d; CG: [$F(1,37) = 557$; $P = 6.7\text{E-}24$]; CAC: [$F(1,37) = 3283$; $P = 9.7\text{E-}38$]). Because CAC sites, in contrast with CGs, are asymmetric on the two DNA strands, we wondered whether this anti-correlation would be different when contrasting gene expression with mCAC levels on its sense or antisense strand (Supplementary Fig. 8). No difference was found, indicating that gene expression is predicted to the same extent by mCAC on either strand, at least for the coverage achieved here. Finally, we used methylseekR to characterize active regulatory sites in the human brain defined as unmethylated (UMR) and lowly methylated (LMR) regions (Supplementary Fig. 9a–e). As observed in other tissues, CG-dense UMRs mostly overlapped with CpG islands and promoter chromatin states (Fig. 1f, Supplementary Fig. 9d, and Supplementary Fig. 10a), while LMRs associated with more diverse states (including enhancers; Supplementary Fig. 1f, and Supplementary Fig. 10a), consistent with their role as distant regulatory sites[31]. Among each LMR and UMR category, significant variations in levels of mCG or mCAC were observed across various chromatin states (Supplementary Fig. 9f). Regarding individual histone marks at LMR and UMR, we further documented specific associations, including patterns of depletion and enrichment specific to UMR shores not characterized previously (see Supplementary Fig. 10b–g for details). Overall, these differences and similarities between mCG and mCAC extend previous results obtained in smaller cohorts of mouse or human samples[29,32].

Regarding histone modifications, while mechanisms mediating their interactions with mCG have been documented, no data are available to describe such a relationship for non-CG contexts. To address this gap, we confronted our consensus model of chromatin states with DNA methylation (Fig. 2e, f). Levels of mCG ([$F(9,324) = 5127$; $P < 0.0001$]) and mCAC ([$F(9,324) = 910.7$; $P < 0.0001$]) strongly differed between states, unraveling previously uncharacterized patterns. First, the lowest levels of mCG were found in the three promoter states (Fig. 2e), corresponding to a strong anti-correlation between DNA methylation and both forms of H3K4me1,3 methylation, consistent with previous findings in other cell types[33]. Accordingly, these three promoter states were defined (Fig. 1e) by high levels of H3K4me3 in combination with either: (i) high H3K4me1 (flanking promoter, Flk-Prom; $P < 0.0001$ for every post hoc comparison, except against the Polycomb repressed state, PcR); (ii) high H3K27ac (active promoter, Act-Prom; $P < 0.0001$ for every comparison against other states), or (iii) intermediate levels of both H3K27ac and H3K4me1 (weak promoter, Wk-Prom; $P < 0.0001$ against other states). In contrast, among these three promoter states, mCAC was particularly enriched in Wk-Prom regions ($P < 0.0001$ against Act-Prom and Flk-Prom; Fig. 2f, Supplementary Fig. 3d). Second, mCG was abundant in transcribed regions defined by either intermediate (weak transcription, Wk-Trans) or high (strong transcription, Str-Trans) H3K36me3. By contrast, mCAC was selectively decreased in the Str-Trans state ($P < 0.0001$ against Wk-Trans). Third, while mCG levels were high in heterochromatin (Heteroch, defined by high H3K9me3), consistent with its role in chromatin condensation, mCAC appeared depleted from these regions ($P < 0.0001$ for every comparison against other states, except PcR and Flk-Prom). These results indicate that interactions between DNA methylation, histones, and chromatin strikingly differ across mCG and mCAC, possibly as a result of brain-specific epigenetic processes in the latter three-letter context[32]. Finally, as expected, ELA did not associate with a global disruption of this cross talk, as no changes in genome-wide levels of mCG ([$F(1,36) = 0.36$;

$P = 0.55$]) or mCAC ([$F(1,36) = 0.07$; $P = 0.80$]) were observed across C and ELA groups for any state.

**Changes in histone marks and chromatin states as a function of ELA.** We investigated local histone adaptations in ELA subjects using diffReps[34]. A total of 5126 differential sites (DS) were identified across the 6 marks (Fig. 3a, b, Supplementary Fig. 11, and Supplementary Data 2) using consensus significance thresholds[35] ($P < 10^{-4}$, FDR-q < 0.1). H3K27ac contributed to 30% of all DS, suggesting a prominent role of this mark. Annotation to genomic features revealed distinct distributions of DS across marks (df = 25, $\chi^2 = 1244$, $P < 0.001$; Supplementary Fig. 12a): H3K4me1- and H3K4me3-DS were equally found in promoter regions and gene bodies, while H3K36me3- and H3K27ac-DS were highly gene-body enriched, and H3K27me3- and H3K9me3-DS found in intergenic/gene desert regions. Sites showing enrichment (up-DS) or depletion (down DS) of reads in ELA subjects were found for each mark, with an increased proportion of down DS associated within H3K4me1, H3K4me3, H3K36me3, and H3K27me3 changes (Supplementary Fig. 12b).

We then used GREAT (Supplementary Data 3), a tool that maps regulatory elements to genes based on proximity, to test whether ELA subjects had histone modifications affecting genes in specific pathways[36]. We performed this GO analysis on each mark and found significant enrichments for three of them (Fig. 3c, d). Importantly, overlaps between enriched GO terms were observed across these three marks: notably, terms related to immune processes, as well as small GTPases and Integrin signaling (Supplementary Fig. 12c) were enriched for H3K36me3- and H3K27ac-DS, suggesting these pathways may play a significant role in ELA.

To strengthen these findings, a complementary analysis was conducted using chromatin state maps[15]. First, we identified genomic regions where a state transition (ST; $n = 61,922$) occurred between groups (Supplementary Data 4). Across the 90 possible ST in our 10-state model, only 56 were observed, with a high proportion (50.2%; * in Fig. 4a) involving regions in quiescent (Quies), Wk-Trans or Enh states in the C group that mostly turned into Quies, Str-Trans, Wk-Trans, and Heteroch states in the ELA group. Furthermore, 17% and 59% of ST occurred in regions within 3 kb of a promoter or in gene bodies (Fig. 4b), respectively, suggesting that ELA-associated changes affected selected chromatin states, and mostly occurred within genes.

We next investigated GO enrichment of ST using GREAT (Fig. 4c, d, Supplementary Data 5) and a co-occurrence score reflecting both the significance of GO terms and their recurrence across multiple ST[35]. Importantly, biological processes (Fig. 4c) with the highest co-occurrence scores were similar to those found from the GO analysis of individual histone marks, and clustered in two main categories: immune system and small GTPases. These terms were significant for ST involving transcription, quiescent, and enhancer states. Regarding molecular functions (Fig. 4d), most enriched categories were related to GTPases, and involved the same types of ST. Therefore, analyses of individual histone marks and chromatin states converged to suggest impairments in similar GO pathways.

**Differential DNA methylation in ELA.** We next sought to identify changes in DNA methylation. As mCG and mCAC were very different, and considering data suggesting possible mCAC-specific processes[28], we used BSmooth[37] to identify DMRs separately in each context, with strictly similar parameters (see "Methods"). DMRs were defined as regions of ≥5 clustered cytosines that each exhibited a significant group difference in

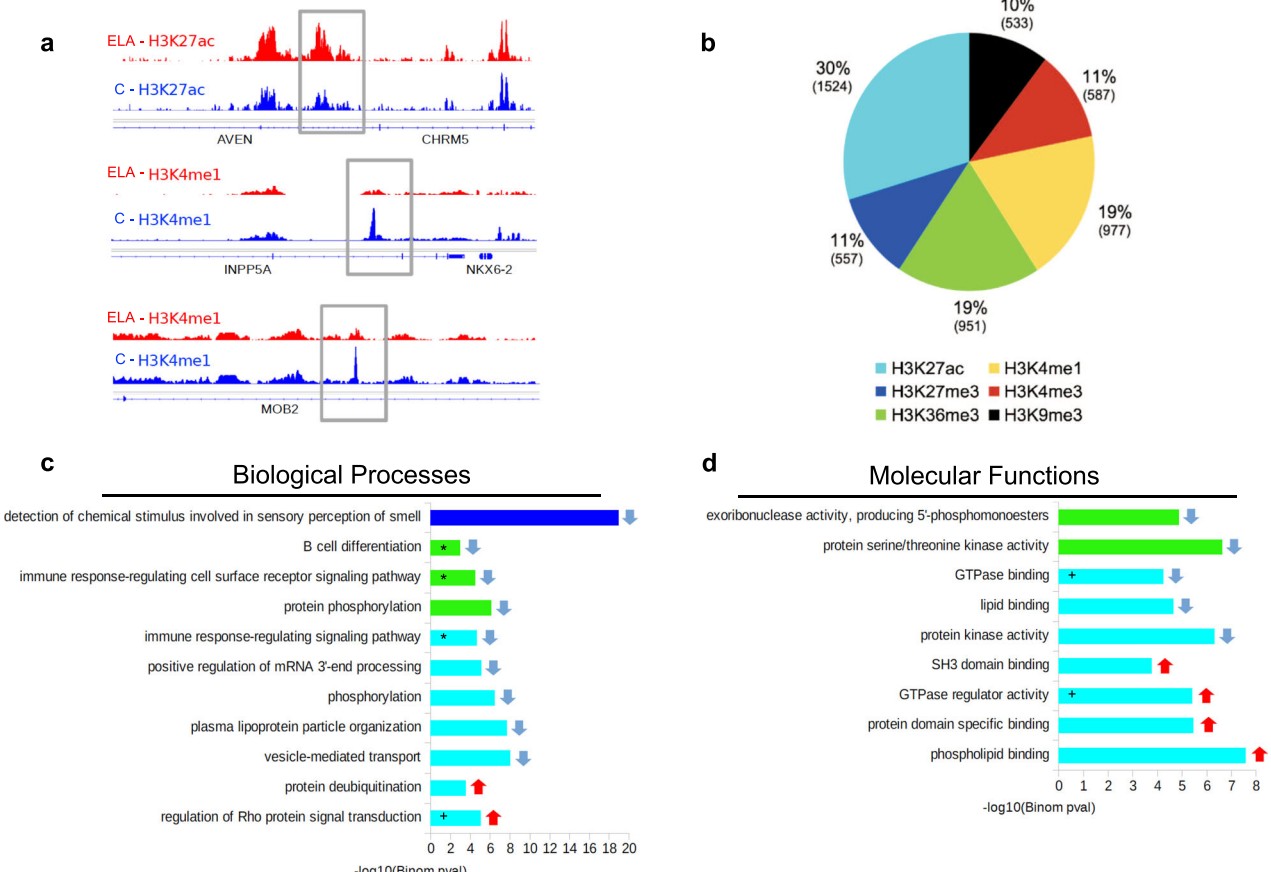

**Fig. 3 Analysis of genomic sites showing differential enrichment for individual histone marks in subjects with a history of early-life adversity (ELA).**
**a** Representation of three top Differential Sites (DS), identified using diffReps[34]. ELA is shown in red, healthy controls (C) are shown in blue. Gray rectangles delineate the coordinates of each DS. **b** Relative proportion of DS contributed by each histone mark. Percentages of the total number of DS, and absolute number of DS (in brackets) are shown for each mark. Both depletion- and enrichment-DS were observed for each of the six marks (Supplementary Fig. 12b). Among genes most strongly affected (Supplementary Table 5), several have been previously associated with psychopathology, such as QKI (H3K27ac top hit)[40,95] or HTR1A (H3K4me3 top hit)[96]. **c**, **d** Top five most significant non-redundant gene ontology "Biological Processes" (**c**) or "Molecular Functions" (**d**) terms enriched for each histone mark DS, as identified by GREAT[36] using hypergeometric and binomial testing (fold change ≥ 1.5 and FDR-q ≤ 0.1 for both tests). Surprisingly, the single most significant result implicated epigenetic dysregulation of odor perception in ELA subjects (consistent with recent clinical studies[97]), while immune processes (indicated by *), and small GTPases (+) were consistently found affected across different marks. Negative logarithmic *P* values are shown for binomial testing. The color indicates histone mark concerned, arrows indicate the direction of event: terms associated with depletion- (down arrow) or enrichment-DS (up arrow). Source data are provided as a Source Data file.

methylation (*P* < 0.001). Also, because age and sex are known to affect DNA methylation[38,39], generalized linear models were computed for each DMR, and only those that remained significant when taking these two covariates into account were kept for downstream analyses. Surprisingly, we found that as many DMRs could be identified in the CAC (*n* = 840) as in the canonical CG (*n* = 795) context, suggesting that cytosines in the CAC context may represent a significant form of plasticity.

While both types of DMRs were similarly abundant and distributed throughout the genome (Fig. 5a, b), they nevertheless showed striking differences. Compared with CG-DMRs, CAC-DMRs were composed of slightly fewer cytosines (Supplementary Fig. 13a, *P* = 2.9E-04) and smaller (Supplementary Fig. 13b, *P* < 2.2E-16). CG-DMRs also affected sites showing a wide range of methylation levels, while CAC-DMRs were located in lowly methylated regions (Fig. 5c, d), consistent with genome-wide lower mCAC levels. In addition, the magnitude of methylation changes detected in the ELA group were less pronounced in the CAC context, with smaller % changes (*P* < 2.2E-16; Fig. 5e, f and Supplementary Fig. 13c) and areaStat values (the statistical strength of DMRs[37]; *P* = 5.8E-08, Supplementary Fig. 13d).

Further strengthening differences between the two contexts, CG- and CAC-DMRs showed no genomic overlap (Supplementary Fig. 13e) and very distinct distributions among UMR and LMR features (Fig. 5g). Finally, when considered collectively, genomic regions where CG-DMRs were identified as a function of ELA showed no group difference in the CAC context (and vice versa for mCG levels at CAC-DMRs; see Fig. 5h), indicating that ELA-related processes do not simultaneously affect both cytosine contexts.

We next characterized genomic features where DMRs occurred and observed that their distribution again strikingly differed (*P* < 2.2E-16; Fig. 6a, b, Supplementary Table 9): CG-DMRs were located in promoters (38.5% in the proximal promoter, promoter1k and promoter3k) and gene bodies (35.4%), while CAC-DMRs were mostly in gene bodies (53%) and intergenic regions (28.1%). Second, we characterized histone modifications around DMRs (Fig. 6c, d, and Supplementary Fig. 14): CG-DMRs were enriched with H3K4me1, H3K4me3 and H3K27ac (Fig. 6c), coherent with our observations that these histone marks (Fig. 1e) and DMRs (Fig. 6a) preferentially located at promoters. In sharp contrast, the two main features characterizing CAC-DMRs were

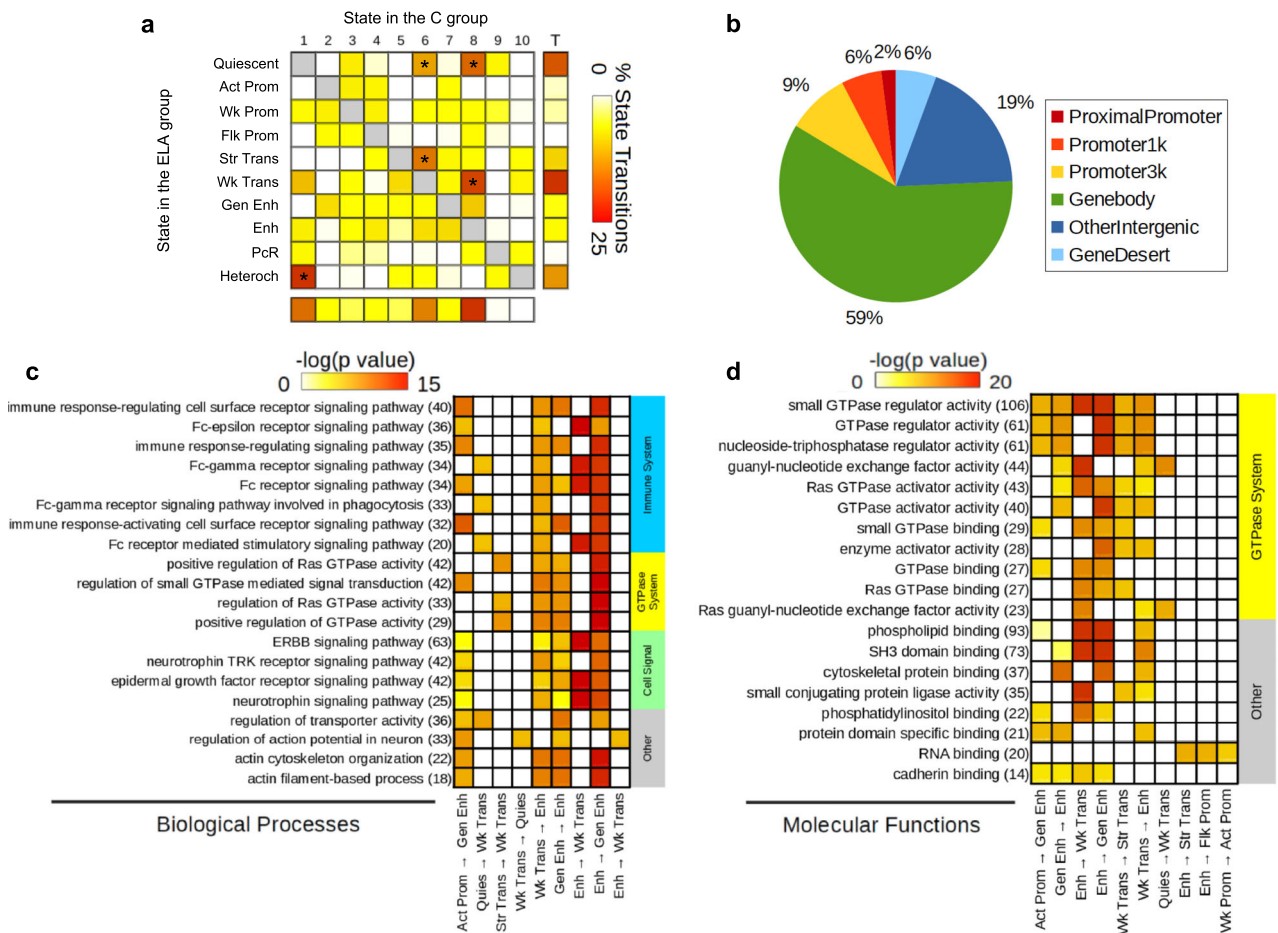

**Fig. 4 Analysis of genomic sites showing a switch between chromatin states as a function of early-life adversity (ELA). a** Percentage of each state transition (ST) type relative to the total number of transitions. For the healthy control (C) versus ELA group comparison, the cumulative percentages of ST from a specific state to any other state are shown in the "Total" and "T" rows/columns. * indicates most frequent STs (see main text). **b** Distribution of ST localizations relative to genomic features, assessed using region_analysis[34] (see "Methods"). **c**, **d** Gene ontology "Biological Processes" (**c**) or "Molecular Functions" (**d**) terms significantly associated with at least three types of ST (for each ST type, each GO term met the following criteria: fold change ≥ 1.5 and FDR-q ≤ 0.1, for both hypergeometric and binomial tests). Terms are grouped based on the overall system involved and ranked by co-occurrence score (in parentheses after each term), which reflects both their significance and their recurrence across multiple ST (see main text and ref. [35]). Individual binomial *P* values for each type of ST and each term are shown by the color gradient. Immune-related and small GTPase terms were most strongly affected, across multiple ST. Of note, a complementary GREAT pathway analysis using MSigDB further strengthened these findings by revealing recurrent enrichment of the integrin signaling pathway (across six types of ST, as well as for H3K27ac down DS; see Supplementary Fig. 12c), which is known to interact extensively with small GTPases[98]. Act-Prom active promoter, Enh enhancer, Flk-Prom flanking promoter, Heterochr heterochromatin, PcR polycomb repressed, Str-Enh strong enhancer, Str-Trans strong transcription, Wk-Prom weak promoter, Wk-Trans weak transcription. Source data are provided as a Source Data file.

an enrichment in H3K36me3 and a depletion in H3K9me3 (Fig. 6d and Supplementary Fig. 14d, f). These differences were further supported by the analysis of chromatin states (*P* < 2.2E-16; Fig. 6e, Supplementary Fig. 14g, and Supplementary Table 10). CAC-DMRs were largely absent from promoter (Act-Prom, Flk-Prom, and Wk-Prom) and enhancer (Str-Enh and Enh) states that were all defined, to varying degrees, by the three marks that primarily characterize CG-DMRs: H3K4me1, H3K4me3, and H3K27ac (Fig. 1e). In addition, CAC-DMRs were (i) enriched in the Wk-Trans state, defined by the presence of H3K36me3, and (ii) depleted from the two states (PcR, Heteroch) characterized by H3K9me3.

Finally, we conducted a GREAT analysis of GO terms enriched for DMRs: CG-DMRs notably associated with terms related to the regulation of neuronal transmembrane potential (Fig. 6f and Supplementary Data 6), in agreement with histone results (Fig. 4c), while CAC-DMRs were enriched for terms related to glial cells (Fig. 6g), consistent with the immune dysregulation

previously observed with histone DS and ST. Altogether, while ELA associates with similar numbers of mCG and mCAC adaptations, these two types of plasticity occur in genomic regions characterized by different histone marks, chromatin states, and GO categories, possibly reflecting the implication of distinct molecular mechanisms.

**Differential gene expression in ELA and combined GO analyses.** Analyses of histones and DNA methylation identified GO terms consistently affected in ELA individuals. To determine how these epigenetic adaptations may ultimately modulate amygdalar function, we characterized gene expression in C (n = 17) and ELA (n = 21) groups using RNA-Sequencing. Samples with similar RNA integrity across groups were sequenced at >50 million reads/sample (Supplementary Fig. 15). Quantification of gene expression was conducted using HTSeq-count[40] and validated by an alternative pseudo-alignment approach, Kallisto[41],

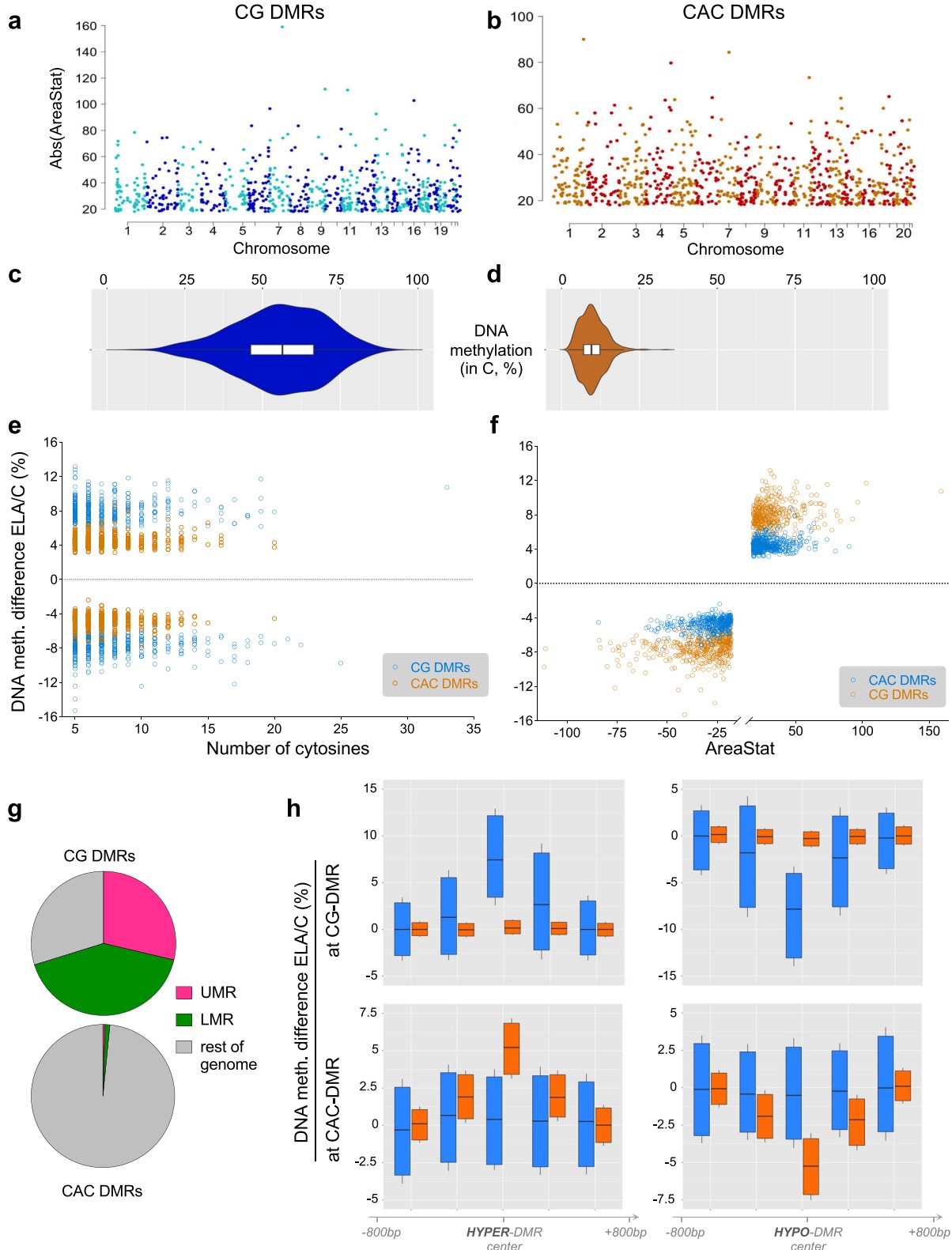

generating very similar results ($r = 0.82$, $P < 2.2\text{E-}16$; Supplementary Fig. 16a). A differential expression analysis between groups was then performed using DESeq2 (Supplementary Data 7). Similar to our epigenetic analyses, we searched for patterns of global functional enrichment, using GO and Gene Set Enrichment Analysis (GSEA)[42]. Enrichment of GO categories using genes that showed nominal differential expression in the ELA group ($P < 0.05$, $n = 735$, Fig. 7a, Supplementary Data 8)

identified numerous terms consistent with previous analyses at the epigenetic level, including immune and small GTPase functions (Fig. 7b). We also used GSEA[42], which does not rely on an arbitrary threshold for significance, and takes the directionality of gene expression changes into account. GSEA identified 163 genome-wide significant sets, among which 109 were related to immune processes and negatively correlated with ELA (Supplementary Data 9, Fig. 7c, d, Supplementary Fig. 16d, e). Therefore,

**Fig. 5 Differential DNA methylation in the CG and CAC contexts in subjects with a history of early-life adversity (ELA). a, b** Manhattan plots of differentially methylated regions (DMR) identified using the BSmooth algorithm in the CG and CAC contexts, comparing control (C) and ELA groups. DMRs were identified separately in each context using the BSmooth algorithm[37], with strictly similar parameters (see "Methods"). They were defined as regions of ≥5 clustered cytosines that each exhibited a significant difference in methylation ($P < 0.001$) and an absolute methylation difference ≥1% between groups. Surprisingly, as many DMRs were identified in the CAC context ($n = 840$) as in the canonical CG context ($n = 795$). **c, d** Methylation abundance in the C group in regions where DMRs were identified in the CG and CAC contexts ($n = 840$ CAC-DMRs, $n = 795$ CG-DMRs). CG-DMRs affected genomic sites showing a wide range of methylation levels (mean ± sem = 55.3 ± 0.5%), while CAC-DMRs occurred in lowly methylated regions (mean ± sem = 10.0 ± 0.1%), resulting in significantly different distributions (Mann–Whitney $U$ test: $U = 686$; $P < 0.0001$). Box and violin plots show median and interquartile range (IQR), with whiskers representing 1.5 IQR. **e** DNA methylation differences observed in ELA subjects compared to the C group in CG- and CAC-DMRs, as a function of the number of cytosines composing each DMR. **f** DNA methylation differences observed in ELA subjects compared to the C group in CG- and CAC-DMRs, as a function of areaStat values, the measure of statistical significance of each DMR implemented by BSmooth. **g** Distinct distributions (chi-square test: $\chi^2 = 884.3$, df = 2, $P < 1E-15$) of CG- and CAC-DMRs among lowly methylated (LMR) and unmethylated (UMR) regions. **h** DNA methylation levels observed in CG and CAC contexts at DMRs ($n = 840$ CAC-DMRs, $n = 795$ CG-DMRs; raw, unsmoothed values) and flanking regions (+/− 1.2 kilobases, kb). No average difference in mCAC levels was observed among C and ELA groups at CG-DMRs (upper panels) that showed either increased (HYPER-DMR, left panels) or decreased (HYPO-DMR, right panels) levels of methylation in ELA subjects (and vice versa for CAC-DMRs, lower panels). Box plots show median and interquartile range, with whiskers representing 0.1 IQR. Source data are provided as a Source Data file.

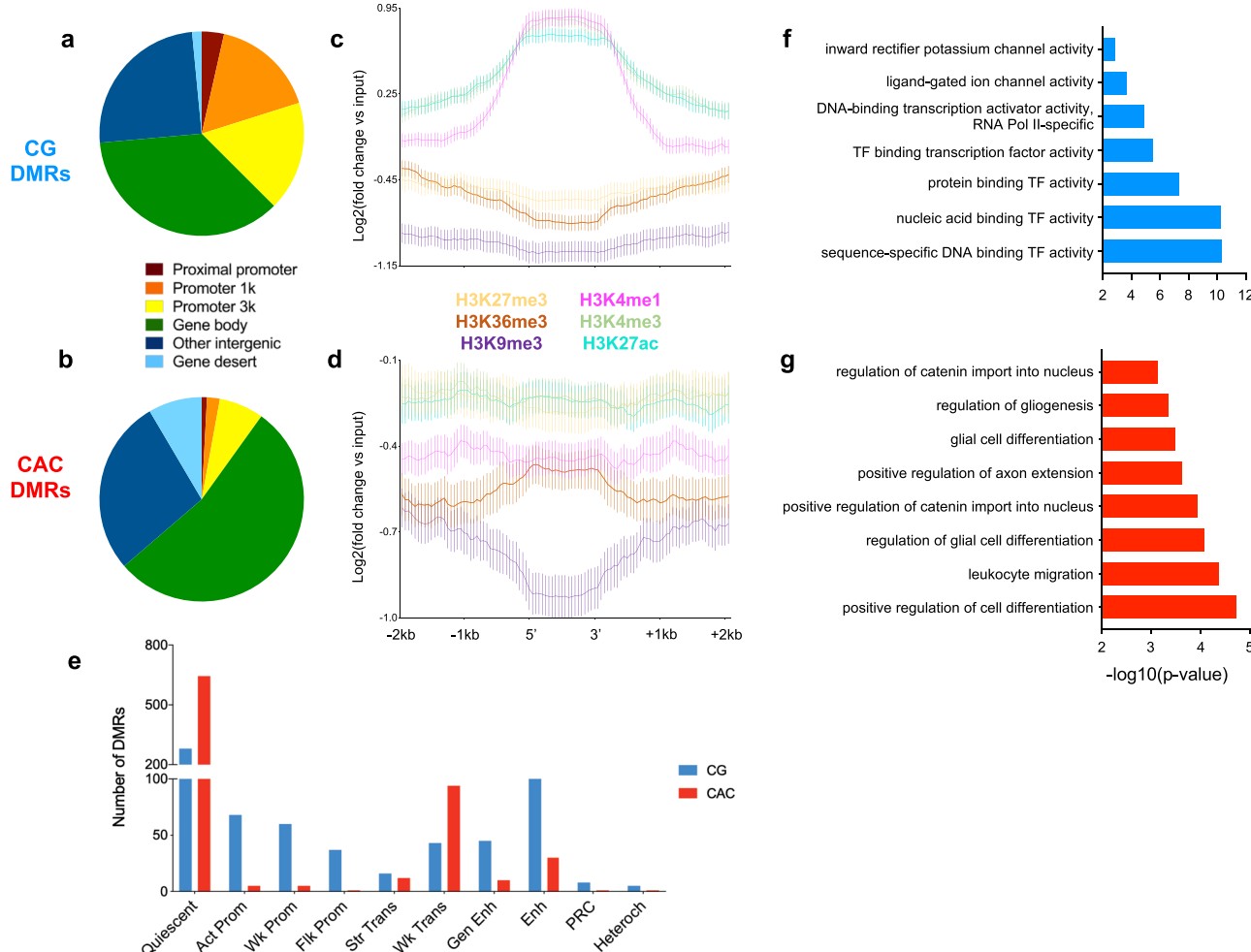

**Fig. 6 Individual histone marks and global chromatin states defining genomic regions where early-life adversity (ELA) associated with differential DNA methylation. a, b** Localization of differentially methylated regions (DMR) in genomic features, identified using region_analysis[34]. Distributions were strongly different among CG and CAC contexts (chi-square test: $\chi^2 = 221.2$, df = 6, $P < 2.2E-16$). **c, d** Histone modifications measured at the level of DMRs and their flanking regions (+/− 2 kilobases, kb). Distributions were very distinct between CG- and CAC-DMRs, with significant interactions between cytosine context and cytosine position along DMRs, for each of the six marks (two-way repeated-measures ANOVA interactions, $P < 0.0001$ for all; see also Supplementary Fig. 14a). Values are mean ± sem. **e** Chromatin states found at DMRs. Similarly, CG- and CAC-DMRs occurred in very different chromatin states (chi-square test: $\chi^2 = 390.4$, df = 9, $P < 2.2E-16$). **f, g** Gene Ontology analysis of CG- and CAC-DMRs using GREAT[36] (see main text). Act-Prom active promoter, Enh enhancer, Flk-Prom flanking promoter, Heterochr heterochromatin, PcR polycomb repressed, Str-Enh strong enhancer, Str-Trans strong transcription, Wk-Prom weak promoter, Wk-Trans weak transcription. Source data are provided as a Source Data file.

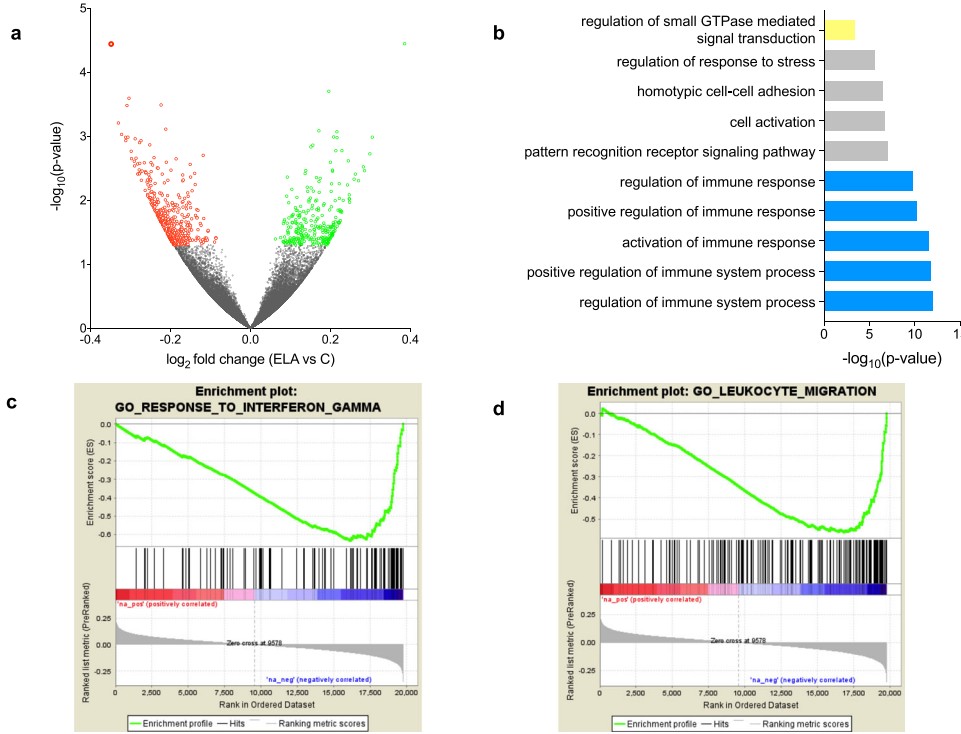

**Fig. 7 Differential gene expression in subjects with a history of early-life adversity (ELA). a** Volcano plot of RNA-Seq data showing the 261 and 474 genes that were up- (green circles) or downregulated (red circles) in the ELA group compared with the control (C) group (GLM model using gender, age, pH, PMI, and RIN as covariates; nominal $P$ value < 0.05). **b** Gene Ontology analysis of the 735 differentially expressed genes in the ELA group. Terms showing evidence of enrichment for differential methylation, histone profile or chromatin state are shown in yellow (small GTPase) or blue (immune processes; see also Supplementary Fig. 16f). **c, d** Gene Set Enrichment Analysis (GSEA) of gene expression changes in ELA subjects. Genes were ranked based on $\log_2$ fold changes from the C versus ELA differential expression analysis ("Ranked list metric", in grey in the lower portion of each panel). Genes with the highest positive fold changes (in red, upregulated in the ELA group) are at the extreme left of the distribution, and those with the lowest negative fold changes (in blue, downregulated in the ELA group) are at the extreme right. A running enrichment score (green line, the upper portion of each panel) was computed for gene sets from the MSigDB curated molecular signatures database and used to identify enriched gene sets[42]. Among the numerous gene sets related to the immune function that showed evidence of genome-wide significant negative correlation with ELA (see main text, and Supplementary Table 14), two representative gene sets are shown (with the middle portion of each panel showing vertical black lines where members of the gene set appear in the ranked list of genes): **c** "Interferon-gamma", and **d** "Leukocyte migration". Of note, an oligodendrocyte-specific gene collection, which we recently found downregulated in the anterior cingulate cortex of subjects with a history of ELA[40], positively correlated with ELA in the amygdala (see Supplementary Fig. 16d), suggesting opposite adaptations in this glial population between cortical and subcortical structures. Source data are provided as a Source Data file.

transcriptomic data revealed gene pathways that in part overlap with those identified using histone marks and DNA methylation.

Because epigenetic and transcriptomic patterns determine and reflect cellular identity, adaptations associating with ELA in the present work may stem from changes in the cellular composition of the amygdala. To explore this possibility, we deconvoluted our bulk tissue measures of gene expression and DNA methylation using BSEQ-sc[43] and the CIBERSORT[44] algorithm. For gene expression, we used as reference single-nucleus transcriptomes recently generated by our group using cortical tissue[45]. Results showed proportions of excitatory (80%) and inhibitory (20%) neuronal subtypes consistent with expectations (Supplementary Fig. 17a) while, importantly, no changes in abundance of neuronal populations, microglia, astrocytes, or oligodendrocytes could be identified as a function of ELA in these analyses (Supplementary Fig. 17b, c). For DNA methylation, we used as reference single-cell non-CG methylomes recently published[46] and found some convergence with cellular estimates generated using RNA-Sequencing (Supplementary Fig. 17e). Again, estimated proportions of different classes of excitatory and inhibitory neurons were unchanged across C and ELA

groups (Supplementary Fig. 17d), reinforcing the hypothesis that ELA-related adaptations reflect changes in cellular phenotypes rather than abundance.

To combine analyses conducted for histones, chromatin states, DNA methylation, and gene expression, we finally grouped GO terms enriched at each level to identify biological mechanisms most consistently affected (Supplementary Fig. 16f). Overall, a clear pattern emerged whereby the highest number of genome-wide significant terms ($n = 101$ GO terms) were related to immune processes, with contributions from each of the four types of data. Second came terms related to small GTPases, which were documented by histone modifications, chromatin states, and gene expression ($n = 24$), followed by terms related to neuronal physiology ($n = 19$, mostly linked with neuronal excitability and sensory processing; Supplementary Data 10), cellular adhesion ($n = 13$), and the cytoskeleton ($n = 5$). Altogether, these combined analyses defined major epigenetic and transcriptomic pathways affected by ELA in the lateral amygdala.

Finally, we sought to determine whether molecular changes associated with ELA in the lateral amygdala might also affect other brain structures implicated in emotional regulation. To do

so, we took advantage of gene expression RNA-Seq data recently published by our group using anterior cingulate cortex (ACC) tissue from a cohort of C and ELA individuals ($n = 50$) that, importantly, included all subjects from the present amygdala study. To identify patterns of shared transcriptional adaptations associated with ELA in both regions, we used Rank–Rank Hypergeometric Overlay (RRHO2[47]). Results uncovered strongly significant overlapping groups of genes that were either commonly downregulated ($P$-adj $= 10^{-487}$, Benjamini–Yekutieli), or commonly upregulated ($P$-adj $= 10^{-388}$), in both the lateral amygdala and ACC (Supplementary Fig. 16b). Strikingly, enrichment analyses (see Supplementary Fig. 16c and full results in Supplementary Data 11) showed that a large majority of GO terms previously identified during the multi-epigenetic investigation of the single amygdala dataset were also recovered by this combined analysis of transcriptomes from two distinct brain regions. Future studies will be necessary to better understand whether similar or divergent epigenetic processes underlie such common transcriptional effects across various brain regions as a function of ELA.

## Discussion

Imaging studies[3] have consistently demonstrated that ELA associates with impaired function of the amygdala. Here, going beyond previous studies[9], we conducted a comprehensive analysis of its potential molecular consequences in this brain region across multiple transcriptional and epigenetic mechanisms. Below, we discuss the implications of our results: first, in the healthy brain; second, in relation to ELA.

Over the last few years, the significance of non-CG methylation, and the possibility that it may fulfill biological functions, have been supported by several lines of evidence, including (i) distinct methylation patterns shown to preferentially affect CAG sites in embryonic stem cells, or CACs in neuronal and glial cells[11], (ii) higher abundance of non-CG methylation in long genes in the human brain[27], and (iii) specific binding of the methyl-CpG-binding domain protein Mecp2 to both mCG and mCAC in the mouse brain[28,48]. Here, we provide additional evidence reinforcing this notion and found that mCG and mCAC exhibit distinct profiles across genomic features and chromatin states, which extends on interactions previously identified in other tissues[49,50], or in the brain for mCG[49]. First, among the three chromatin promoter states (Act-Prom, Wk-Prom, Flk-Prom, see Fig. 2e, f), mCAC was selectively enriched in Wk-Prom, which was not observed for mCG. Considering that Wk-Prom was relatively depleted in H3K27ac and H3K4me1 compared to the two other promoter states, it is possible to hypothesize that these two histone modifications may potentially repress mCAC accumulation in brain tissue. A second dissociation consisted in the fact that lower mCAC levels were measured in Str-Trans compared with Wk-Trans regions, while no such difference was observed in the CG context. This may result at least in part from higher levels of H3K36me3 observed in the Str-Trans state. Third, among the two tightly compacted chromatin states defined by the repressive mark H3K9me3, PcR and Heteroch, the latter state was characterized by higher DNA methylation in the CG, but not in the CAC, context, as well as by a relative increase in H3K9me3 and a decrease in H3K27me3. While there is currently no data, to our knowledge, supporting a potential interaction between H3K27me3 and non-CG methylation[11], a role for H3K9me3 can be speculated considering studies of cellular reprogramming. Indeed, in vitro dedifferentiation of fibroblasts into induced pluripotent stem cells associates with the restoration of non-CG methylation patterns characteristic of stem cells, except in genomic regions characterized by high levels of H3K9me3[51]. Therefore, the possibility exists that H3K9me3 may be implicated in the regulation of mCAC in the brain, a hypothesis that warrants further investigation.

Beyond molecular interactions in physiological conditions, this study was primarily designed to investigate molecular consequences of ELA. Over the last two decades, considerable evidence has associated enhanced inflammation with stress-related phenotypes such as depression, in particular, based on measures of cytokines and inflammatory factors in blood samples[52]. Limited molecular data, however, document how this pro-inflammatory state may translate in the brain. Available studies focused on cortical structures of the frontal lobe and reported conflicting results for the expression of related genes[53–55]. At the histological level, while the prevailing view holds that stress-related psychopathology associates with tissue inflammation[52], studies conducted on the amygdala showed discordant results, with lower densities of glial cells in some[56,57] but not all[58] studies. In this work, integration of genome-wide data on DNA methylation, histone, and gene expression found converging evidence for significant enrichment in immune-related GO terms (Figs. 3, 4, 6, 7, and Supplementary Fig. 17). This included decreased expression of genes encoding the complement system, Toll-like receptors, clusters of differentiation, and the major histocompatibility complex, altogether arguing for a meaningful contribution to psychopathological risk. Of note, deconvolutions of transcriptomic or methylomic data provided no indications of changes in amygdala cellular composition as a function of ELA, suggesting that the reported molecular adaptations may reflect decreased activity rather than impaired recruitment or proliferation of microglial and astrocytic cells, the main immune actors in the brain. Altogether, our data suggest that dysregulation of immune-related processes in the amygdala may play an important role in long-term consequences of ELA and in the pathophysiology of depression.

While proteins from immune pathways have been historically identified and studied in the context of immune function and associated circulating cells (eg lymphocytes, monocytes, and granulocytes), a growing and significant literature now indicate that they are also largely expressed by neuronal cells, and play important role in the regulation of synaptic plasticity[59–64]. Consistently, other pathways most significantly altered in ELA subjects were related to small GTPases, a large family of GTP hydrolases that regulate synaptic structural plasticity, notably through interactions with the cytoskeleton[65]. The association observed for small GTPases was also accompanied by changes affecting GO terms related to the cytoskeleton. Overall, our findings, therefore, point toward altered synaptic plasticity in the lateral amygdala in relation to ELA and depression and reveals part of underlying epigenetic mechanisms at DNA methylation and histone levels. While few molecular studies in humans previously documented this hypothesis[66], it strongly resonates with the wealth of human imaging and animal data that shows structural and functional plasticity in this brain region as a function of stressful experiences[4].

Finally, we wondered whether mCAC may also contribute to molecular responses to ELA in the brain. We found that similar numbers of differential methylation events could be detected across CAC and CG contexts in ELA subjects, suggesting that both contexts might be sensitive to behavioral regulation. While previous studies already showed that ELA associates with widespread effects on mCG throughout the genome, they were conducted using methodologies primarily designed for the investigation of mCG (methylated DNA immunoprecipitation coupled to microarrays[67], reduced representation bisulfite sequencing[40]). In comparison, the present WGBS study provides a more comprehensive and unbiased assessment of the overall

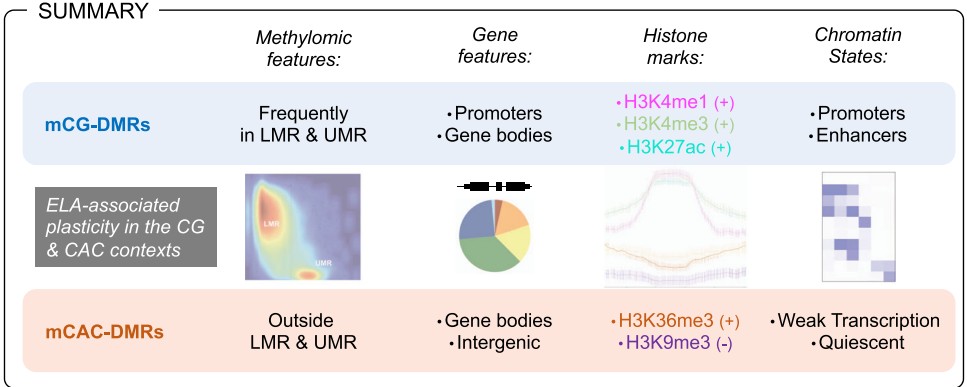

**Fig. 8 Methylomic adaptations associated with early-life adversity (ELA) in the CG and CAC contexts show multiple distinct properties.** The figure depicts a summary of methylomic and gene features, as well as histone marks and chromatin states that characterize genomic sites where differentially methylated regions (DMR) were identified as a function of ELA in the CG and CAC contexts. See main text for details. LMR lowly methylated regions, UMR unmethylated regions, + and − indicate enrichment and depletion in the corresponding histone mark, for each DMR category.

methylome, and represents, to our knowledge, the first indication in humans that the mCAC form of DNA methylation might be affected by ELA and related depressive phenotypes. This is consistent with recent mouse work[68] that provided evidence for an effect of early-life positive experiences (in the form of environmental enrichment during the adolescence period) on non-CG methylation, suggesting that both beneficial and detrimental experiences may modulate this noncanonical epigenetic mechanism. Importantly, our combined investigation of DNA methylation and histone marks provides further characterization of this form of plasticity. Strikingly, mCAC and mCG changes occurred in genomic regions that appeared distinct at every level of analysis (see summary in Fig. 8), including genic or methylomic features, individual histone marks, chromatin states, and GO categories. Accordingly, CG-DMRs primarily located among promoter regions and gene bodies; they were enriched in H3K4me1, H3K4me3, and H3K27ac, and present across all chromatin states, but mostly in Prom and Enh. In comparison, CAC-DMRs were less frequently found in promoters, enriched in H3K36me3 and depleted in H3K9me3, and mostly associated with Quiescent and Wk-Trans chromatin states. Furthermore, previous studies on non-CG methylation focused on the comparison of distinct cell types (glial *vs* neuronal cells[12], or excitatory vs inhibitory neurons[24]), and identified parallel and significant differences in both CG and non-CG contexts at common genomic sites. In sharp contrast, regions showing differential methylation as a function of ELA in the present study were clearly context-specific (Fig. 5h and Supplementary Fig. 13e), indicating a more subtle and specific modulation of DNA methylation by ELA than by cell identity. Overall, these results are consistent with a model whereby the cascades of neurobiological adaptations associated with ELA result from and contribute to distinct pathophysiological phenomena that differentially manifest at the level of CG and CAC sites. It is possible also to speculate that part of these adaptations may result from the impact of ELA on mechanisms that drive the developmental emergence of mCAC. Along this line, a molecular pathway has recently started to be unraveled in the mouse: the methyltransferase Dnmt3a was shown to mediate the progressive postnatal accumulation of DNA methylation in the CA context[13], while in vivo recruitment of MeCP2 primarily relies on mCG and mCAC levels (rather than methylation at other contexts, including CAT, CAA, or CAG[28]). These 2 studies suggest that DNMT3a and MeCP2 may be implicated in human in the particular cross-talk that emerges during brain maturation between

mCAC and specific histone modifications (H3K27ac, H3K4me1, H3K36me3, and H3K9me3). Therefore, future investigations should focus on these marks, and related histone-modifying enzymes, to better decipher the lifelong impact of ELA on molecular epigenetic interactions.

Of note, this study has limitations. First, due to technical constraints at the beginning of the project, pools of amygdala tissue from several subjects were analyzed for ChIP-Seq, while RNA-Seq and WGBS were conducted separately for each sample. While this may have affected our results (for example, by obscuring subtle subject-specific histone changes not detected at pool level), the convergence of functional annotations observed across multiple types of data suggests a modest detrimental impact of the pooling approach. Second, the field of genomics is currently moving towards the molecular analysis of single-cells, with the hope of achieving higher resolution and a better understanding of psychopathology. In comparison, this work focused on bulk tissue only and, as such, may have missed epigenetic processes affecting individual or rare cell types. Finally, all ELA subjects died during a major depressive episode by means of suicide. Therefore, it is possible that part of the molecular adaptations that we uncovered, and cautiously associate with ELA, derive from these complex phenotypes rather than stem specifically from ELA. Exploring this hypothesis will require replication in larger cohorts and additional clinical groups (eg, depressed suicides with no history of ELA).

In conclusion, the epigenetic and transcriptomic landscape of the lateral amygdala exhibit targeted reconfigurations as a function of ELA. This reprogramming can be detected consistently across multiple epigenetic mechanisms, including the newly recognized form of DNA methylation affecting CAC sites. Future studies will hopefully define the extent to which non-CG methylation at CACs, and potentially at other cytosine contexts, contribute to the adaptive and maladaptive encoding of life experiences in the brain.

## Methods

**Human samples and tissue dissections.** Postmortem lateral amygdala brain tissue was obtained in collaboration with the Quebec Coroner's Office, from the Douglas-Bell Canada Brain bank (douglasbrainbank.ca/, Montreal, Canada). This study included (i) subjects who died suddenly without prolonged agonal state or protracted medical illness, and with no history of psychiatric disorder (Controls, C, $N = 17$), and (ii) subjects with a history of severe child abuse, who died by suicide in the context of a major depressive episode (Early-life adversity, ELA, $N = 21$). Sample characteristics are presented in Supplementary Table 1, while the type of abuse and mean of death are detailed in Supplementary Table 2. Groups were

matched for age, postmortem interval (PMI), and brain pH. Psychological autopsies were performed by trained clinicians on both controls and cases, with the informants best-acquainted with the deceased, as described previously[69] and as validated by our group and others[70–75]. Diagnoses were assigned based on DSM IV criteria. Characterization of early-life histories was based on adapted Childhood Experience of Care and Abuse (CECA) interviews assessing experiences of sexual and physical abuse, psychological abuse, as well as neglect[76,77], and for which scores from siblings are highly concordant[70,77]. We considered as severe early-life adversity reports of non-random major physical and/or sexual abuse during childhood (up to 15 years). Only cases with the maximum severity ratings of 1 and 2 were included. This information was then complemented with medical charts and coroner records. Ethical approval was obtained from the Institutional Review Board of the Douglas Mental Health University Institute (REB#08-14). Written informed consent was obtained from the families of each of the deceased subjects prior to inclusion in the study.

**Next-generation sequencing**. WGBS, RNA-Seq, and ChIP-seq experiments were carried out by expert technicians at the McGill University and Genome Quebec Innovation Center, following standard operating procedures from the International Human Epigenome Consortium (IHEC, see ihec-epigenomes.org/).

**ChIP-seq library preparation**. The following antibodies were used for chromatin immunoprecipitation: (i) H3K4me1: Cell Signaling Technologies, cat #5326BF, lot #2 (quantity: 3 µg/concentration: 0.015 µg/µl); (ii) H3K4me3: Cell Signaling Technologies, cat #9751BF, lot#6 (quantity: 5 µg/concentration: 0.025 µg/µl); (iii) H3K9me3: Abcam, cat #Ab8898, lot #GR93671-1 (quantity: 3 µg/concentration: 0.015 µg/µl); (iv) H3K27me3: Cell Signaling Technologies, cat #9733S, lot #6 (quantity: 10 µg/concentration: 0.05 µg/µl); (v) H3K27ac: Diagenode, cat #pAB-196-050, lot #A1723-0041D (quantity: 6 µg/concentration: 0.03 µg/µl); (vi) H3K36me3: Active motif, cat #MABI0333, lot #12003 (quantity: 2 µg/concentration: 0.01 µg/µl). Libraries were prepared using the automated protocol for the Kapa HTP Library Preparation Kit (Illumina), and sequencing was performed using the Illumina HiSeq 2000, as per the manufacturer's instructions, to achieve at least 30 and 60 million reads for narrow (H3K27ac and H3K4me3) and broad (H3K27me3, H3K36me3, H3K4me1, and H3K9me3) marks, respectively (Supplementary Fig. 1a).

**ChIP-seq data processing**. Trimmomatic[78], BWA[79], Picard, and deepTools[80] were used to pre-process and align the sequencing reads. Global visualization for the ChIP-seq data was accomplished using IGV[81] and ngs.plot[82]. Inter-sample correlations and hierarchical clustering were achieved using deepTools. Identification of differential enrichment sites for each histone mark was done using diffReps with window size 1000 bp, sliding step 100 bp, and fragment size 200 bp[34]. A FDR < 10% and P < 0.0001 for the negative binomial tests were used as significance cutoffs. ChromHMM was used to partition the genome into 200-bp bins and annotate them to chromatin states[15]. A 10-state model was chosen and applied to all datasets. A consensus map was first created for general characterization of the amygdala chromatin states, using genomic regions with at least 6/11 samples in accordance with a state. For comparisons of the two clinical groups, group-specific maps were defined using regions showing at least a 70% agreement between samples (at least 3/4C pools and 5/7 ELA pools). State transitions (ST) were then defined as regions with differing states between the C and ELA maps. For characterization of the distribution of DS and ST, we used the region_analysis package (Python v3) to annotate them to genomic features[34].

**WGBS library preparation**. Whole-genome sequencing libraries were generated from 700 to 1000 ng of genomic DNA spiked with 0.1% (w/w) unmethylated λ DNA (Promega) previously fragmented to 300–400 base pairs (bp) peak sizes using the Covaris focused-ultrasonicator E210. Fragment size was controlled on a Bioanalyzer DNA 1000 Chip (Agilent) and the KAPA High-Throughput Library Preparation Kit (KAPA Biosystems) was applied. End repair of the generated dsDNA with 3′- or 5′-overhangs, adenylation of 3′-ends, adaptor ligation, and clean-up steps were carried out as per KAPA Biosystems' recommendations. The cleaned-up ligation product was then analyzed on a Bioanalyzer High Sensitivity DNA Chip (Agilent) and quantified by PicoGreen (Life Technologies). Samples were then bisulfite-converted using the Epitect Fast DNA Bisulfite Kit (Qiagen), according to the manufacturer's protocol. Bisulfite-converted DNA was quantified using OliGreen (Life Technologies) and, based on quantity, amplified by 9–12 cycles of PCR using the Kapa Hifi Uracil+DNA polymerase (KAPA Biosystems), according to the manufacturer's protocol. The amplified libraries were purified using Ampure Beads and validated on Bioanalyzer High Sensitivity DNA Chips, and quantified by PicoGreen. Libraries were run on an Illumina HiSeq 2000 (100 bp paired end), yielding ≈164 million reads/library on average (Supplementary Fig. 5c), and generating around 6.2 billion reads in total across the whole cohort.

**WGBS data processing**. As previously described[83], methylome libraries were aligned using BWA 0.6.1[79] after converting all the reads in bisulfite mode to the human hg19/GRCh37 genome reference. Both reads in a pair were trimmed of any low-quality sequence at their 3′-ends (with Phred scale score > = 30). Post-process

read mappings were made as previously described[83], including clipping 3′-ends of overlapping read pairs in both forward and reverse strand mappings, filtering duplicate, low-mapping quality reads, read pairs not mapped at the expected distance based on the library insert size as well as reads with >2% mismatches. Methylation calls of individual cytosines in both CG and CAC contexts were extracted using Samtools in mpileup mode. Cytosine overlapping SNPs from dbSNPs (137) and CpGs located within ENCODE DAC blacklisted regions or Duke excluded regions[84] were discarded.

**DNA methylation data characterization**. All analyses conducted to characterize the genome-wide abundance and distribution of CG and CAC methylation were done by focusing on cytosines showing a coverage > = 5 (Fig. 2 and Supplementary Figs. 6–9). We used the region_analysis package[34] to assign each cytosine to a genomic feature (Supplementary Fig. 7), using the Ensembl v75 annotation for consistency with RNA-Sequencing data analysis (see below). MethySeekR was used to call CpG-rich, unmethylated regions (UMR), as well as CpG-poor low-methylated regions (LMR), as described by Burger et al.[85].

**Differential methylation analysis**. Differential methylation analysis was conducted using BSmooth, as described previously[86]. The context of each C was determined, which allowed us to classify each C of the genome as CG or CAC. Methylation levels for each site were estimated by counting the number of reported C ("methylated" reads) divided by the total number of reported C and T ("methylated" plus "unmethylated" reads) at the same position of the reference genome. To identify differentially methylated regions in the CG context, we performed a strand-independent analysis of CG methylation where counts from the two Cs in a CG and its reverse complement (position $i$ on the plus strand and position $i + 1$ on the minus strand) were combined and assigned to the position of the C in the plus strand. The summarized methylation estimates of strand-merged CG sites from the 21 ELA and 17 control samples were used to identify differences in methylation, using the R package BSmooth/BSseq[37] at default parameters. To minimize the noise in methylation estimates due to low-coverage data, we restricted the differential methylation analysis to CpG sites with coverage ≥4 sequence reads in at least ten samples in each condition, which still allowed us to interrogate changes in methylation levels at ~18 million CG and ~39 CAC million sites. The same strategy was applied for differential methylation analysis in the CAC context, except that by definition methylation data originated for each CAC site from one DNA strand only. We identified differentially methylated regions (DMRs) as regions containing at least five consecutive CG, or CAC, sites that were significantly differentially methylated using an unpaired Welch $t$ test (P < 0.001) and that exhibited at least a 1% difference in mean methylation levels between ELA and C groups. To rule out potentially confounding effects of age and sex (two factors known to contribute to variations in DNA methylation[38,39]), a generalized linear model taking into account these two variables was computed on mean methylation levels for each CG- and CAC-DMR. Only those DMRs for which differential methylation between C and ELA subjects remained significant when correcting for age and sex were considered for downstream analyses. Finally, genomic features were attributed to DMRs using the region_analysis package, similar to the annotation of ChIP-Seq DS or ST, while intersections of DMRs with UMRs and LMRs were determined using Bedtools.

**RNA-Sequencing library preparation**. RNA was extracted from homogenized brain samples using the RNeasy Lipid Tissue Mini Kit (Qiagen). The quantity and quality of extracted RNAs were measured using an Agilent 2100 Bioanalyzer. No sample was excluded because of low RIN value. RNA-Sequencing libraries were prepared by expert technicians at the McGill University and Genome Quebec Innovation Center, using IHEC procedures. Briefly, we used the TrueSeq Stranded Total RNA Sample Preparation kit (Illumina), using the Ribo-Zero Gold kit (Illumina) for the depletion of ribosomal RNA, followed by first and second-strand cDNA synthesis and fragmentation of dsDNA. Then, fragmented DNA was used for A-tailing, adaptor ligation, and 12 cycles of PCR amplification. Libraries were quantified using a high sensitivity chip on a Labchip (PerkinElmer), quantitative PCR (KAPA Library Quantification, Kapa Biosystems), and PicoGreen (Life Technologies). Three libraries were run per lane of an Illumina HiSeq 2000 (100 bp paired end), yielding ≈54 million reads/library (Supplementary Fig. 15b).

**RNA-Sequencing alignment, counting, and differential expression analysis**. As described previously[87], we used: FASTX-Toolkit (hannonlab.cshl.edu/fastx_toolkit/links.html) and Trimmomatic[78] for adapter trimming; Bowtie2 for alignment; TopHat[88] for transcript alignment; HTSeq-count[89] or Kallisto[41] for counting; and DESeq2[90] for differential expression analysis. Following high-throughput sequencing, 100 bp paired-end reads were aligned to the hg19 human genome using TopHat v2.1.0 (tophat.cbcb.umd.edu/) with a mate insert distance of 75 bp (-r) and library type fr-unstranded. Reads passing a mapping quality of at least 50 were used for gene and transcript quantification. Gene annotations from the Ensembl release 75 were used for gene-level quantification. We used HTSeq-count version 0.6.1p1 (www.huber.embl.de/users/anders/HTSeq/doc/overview.html), using the intersection-nonempty mode, and results were combined to form a count matrix of 20,893 transcribed RNAs across 50 samples. As an alternative strategy to

HTSeq-count, we also processed reads through the pseudo-aligner Kallisto. Here, expression counts were obtained for isoforms using Kallisto (v. 0.43.0). Then, the txIMport (v. 1.0.3) R package was used to reconstruct gene-level counts using the isoform-level counts generated by Kallisto. For differential expression analysis, genes with no mapped fragments were removed. Furthermore, genes with low counts were removed by keeping only those with at least 20 counts per subject in average. Using HTSeq-count, differential expression analysis was performed using the DESeq2 general linear model (GLM) using the following covariates: gender[91], age[92], pH, PMI, and RIN[93], based on previous literature documenting their impact on human brain RNA-Seq datasets.

**Gene set enrichment analysis (GSEA).** GSEA was performed as previously described[40,42]. $Log_2$ fold changes were obtained for each gene from the differential gene expression analysis. Genes were ranked based on their fold changes where genes with the highest positive fold changes were at the top of the list and those with the lowest negative fold changes were at the bottom of the list. The ranked gene list was then used as an input for the GSEAPreranked tool, with the "classic" enrichment score calculation option selected. The C2 curated gene sets molecular signatures database was used to identify enriched gene sets (family-wise error rate, FWER < 0.1).

**Rank–Rank Hypergeometric Overlay (RRHO2).** RRHO2 was performed as described by Cahill et al.[47], using the corresponding R package: https://github.com/Caleb-Huo/RRHO2. Briefly, two lists of genes present in both the lateral amygdala and ACC RNA-Seq datasets were identified, and the following metric computed for each brain region: $-log_{10}(P value) \times sign(log_2 fold change)$. Then, the RRHO2 function was applied to the two gene lists at default parameters (with step size at 140). The significance of hypergeometric overlaps is reported as $log_{10} P$ values corrected using the Benjamini–Yekutieli procedure. Lists of genes generated by RRHO2, and corresponding to most significant hypergeometric overlaps, were then used for GO enrichment analysis, similar to RNA-Seq data.

**Deconvolution of cellular composition.** To assess the abundance of various cell types in our amygdala samples, we used BSEQ-sc[43] and the CIBERSORT[44] algorithm and applied these to both our RNA-Sequencing and WGBS data. For deconvolution of these data, we used as reference "signatures": (i) for gene expression: a matrix built from single-nuclear RNA-Sequencing data recently generated by our group using prefrontal cortical tissue[45] (archived on GEO Datasets under the reference series GSE144136), and analyzed using unsupervised graph-based clustering[94], and (ii) for DNA methylation: the non-CG methylation matrix generated by Luo et al.[46] using single-nucleus methyl-cytosine sequencing. Relative fractions of cells were computed and are displayed in Supplementary Fig. 17.

**Gene ontology.** We used GREAT v3.0.0[36] to identify the enrichment of gene categories in differential sites (DS) or state transition sites (ST) obtained from ChIP-Seq experiments, and for DMRs from WGBS experiments. DS, ST, and DMRs were associated with genes using the default proximal (5 kb upstream, 1 kb downstream of TSS) and distal (+/− 1 Mb of TSS) definition of regulatory regions. Biological process and molecular function gene categories were kept if they passed both the hypergeometric and binomial tests with a fold enrichment ≥1.5 and FDR Q ≤ 0.1. Significant GO terms with less than five genes associated with ST, DS, or DMRs were discarded. To account for the recurrence of terms across multiple combinations of ST, we calculated a co-occurrence score for each GO term, consisting of the sum of the –log10 of the binomial $P$ value for each ST enriched in this term, as described by Feng et al.[35].

**Reporting summary.** Further information on research design is available in the Nature Research Reporting Summary linked to this article.

## Data availability
Raw and processed data reported in this study using brain tissue from the lateral amygdala and anterior cingulate cortex are publicly available via the Gene Expression Omnibus with accession "GSE151827". All other relevant data supporting the key findings of this study are available within the article and its Supplementary Information files or from the corresponding author upon reasonable request. A reporting summary for this article is available as a Supplementary Information file. Source data are provided with this paper.

## Code availability
The analysis code that supports the main findings of this study is detailed in the Supplementary Information file.

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

## Acknowledgements

P.E.L. was supported by fellowships from the "Fondation Fyssen", the Canadian Institutes of Health Research (201311MFE-320636-218885), the American Foundation for Suicide Prevention (PDF-0-081-13), the "Fondation pour la Recherche Médicale" (ARF20160936006), and by fundings from the Bettencourt-Schueller Foundation, the Brain Canada Foundation, the "Fondation Deniker", the "Fondation de France" (N° Engt:00081244), and the "Union Nationale de Familles et Amis de Personnes Malades et/ou Handicapées Psychiques".

## Author contributions

P.E.L. and G.T. conceived the study. G.G.C., E.M., J.Y., and P.E.L. performed RNA and DNA extractions. T.K. and T.P. provided library preparations and next-generation sequencing through the IHEC consortium, and A.R. prepared ChIP-Seq libraries. M.A.C., J.F.T., and P.E.L. analyzed histone data. Z.A., J.F.T., and P.E.L. analyzed methylation data. L.V.K., J.F.T., and P.E.L. analyzed RNA-Seq data, while the Kallisto analysis was conducted by J.C.G. A.P. and J.F.T. conducted the BSmooth differential methylation analysis. M.A., L.C.V., C.E., I.Y., N.M., and G.T. provided lab resources and equipment. P.E.L., M.A.C., and G.T. prepared the paper, and all authors approved its final version.

## Competing interests

The authors declare no competing interests.
