## [Peer Review File · Nature Communications]

Reviewer #1 (Remarks to the Author):

In this manuscript, the authors present novel and very interesting data from postmortem brain tissue, characterizing genome-wide DNA methylation, as well as 6 different histone marks and the transcriptome using next generation sequencing from the same tissue. The analyses are performed in tissue (bulk) from the lateral amygdala and focuses on exposure to early life adversity and depression for case/control comparisons (total N = 38 brains).

The major strength of this paper is the multi-level epigenetic characterization in the same sample. This reviewer is not aware of any other published manuscript describing such a broad epigenetic characterization of postmortem brain tissue and the interrelationships of these epigenetic features. Especially the focus on non-CG methylation is exciting.

In the context of early life adversity and depression, the strongest findings come from DNA methylation analyses, implicating both CG and CAC methylation as associated with exposure. Also data from several levels of investigation convergently point to immune pathways and small GTPase signaling as being affected by adversity and depression in the amygdala.

Overall, as very interesting paper and great resource for researchers with very solid methodology and thoughtful analyses – with the multilevel epigenetic data being the strongest part. The findings for adversity are not replicated and should therefore be interpreted more cautiously, with the reviewer's understanding that this is a uniquely phenotyped sample in this regard. Could the authors explore, whether at least some of the findings in the amygdala can also be seen in data from other brain regions from the same samples?

I have some minor comments.

Could the authors give a bit more information on the cases, i.e. type of abuse, specific depression diagnosis (recurrent, psychotic etc..) and means of suicide?

Even though pooling of samples for analyses is plausible due to the amount of material necessary for the different ChIPSeq protocols, it is a limitation and should be discussed and directly mentioned in the results when mentioning the sample size and suppl. Tables 1 and 2. Also it should be more clearly stated that WGBS and RNA seq was performed in the individual samples, this difference is not very clear in the main text and should also be mentioned there. For RNA seq, the exact number of samples should be mentioned, and whether any sample was excluded for low RIN, this is not clear but relevant, especially once the data are compared to the pooled sequencing in ChIPSeq.

The discussion should include a limitations section, mentioning the pooling approach, lack of single cell resolution and lack of replication at this point with a more cautious interpretation of the molecular and biological findings with regards to adversity.

Reviewer #2 (Remarks to the Author):

This is a very innovative study on the lateral amygdala of subjects diagnosed with depression/suicide and a history of early life adversity and controls.

While the overall sample size is small (N=21 and N=17 per group), the study is unique because

(i) it is focused on the lateral amygdala, a clinically extremely important brain region but , because of limitations in tissue quantities, barely explored in the field of psychiatric neurogenomics, (ii) 8 different epigenetic markings (6 histone modifications and CAC and CG methylation profiles) are explored plus RNA-seq and (iii) Douglas-Bell Canada brain bank is one of only few (maybe even the only one) brain bank worldwide to have built a collection of brains from subjects for early early life adversity.

So, the study will have a significant impact in the field and will make an extremely important contribution.

My suggestions to the authors:

a) their findings , given that they examined 8 epigenetic markings, are as expected, 'multidimensional', but they should make in the discussion section a better effort to synthesize their findings , or at least provide as the last figure an schematic or illustrative overview summarizing all their main findings so that the Reader does not get overwhelmed with the many details of their findings.

b) technically: the Authors relied on Diffrep analysis tools to study histone case control differences. Which is fine, but keep in mind that Diffrep (if I am correctly informed) does not rely on peak calling algorithms but on sequence windows and may be prone to false positive findings, and thus their paper would benefit from additional analyses with peak calling, not to replace their Diffrep findings but to confirm and extend.

c) there should be some discussion whether and how the pooling of samples from different individuals may or may not have affected the results. While such strategy may be unavoidable given the limited quantity of amygdala tissue , there should be a brief discussion in the Discussion section.

Non-CG methylation and multiple epigenetic layers associate child abuse with immune and small GTPase dysregulation

Rebuttal letter

Reviewer #1

In this manuscript, the authors present novel and very interesting data from postmortem brain tissue, characterizing genome-wide DNA methylation, as well as 6 different histone marks and the transcriptome using next generation sequencing from the same tissue. The analyses are performed in tissue (bulk) from the lateral amygdala and focuses on exposure to early life adversity and depression for case/control comparisons (total N = 38 brains).

The major strength of this paper is the multi-level epigenetic characterization in the same sample. This reviewer is not aware of any other published manuscript describing such a broad epigenetic characterization of postmortem brain tissue and the interrelationships of these epigenetic features. Especially the focus on non-CG methylation is exciting.

In the context of early life adversity and depression, the strongest findings come from DNA methylation analyses, implicating both CG and CAC methylation as associated with exposure. Also, data from several levels of investigation convergently point to immune pathways and small GTPase signaling as being affected by adversity and depression in the amygdala.

We thank the reviewer for the very positive comments.

Overall, a very interesting paper and great resource for researchers with very solid methodology and thoughtful analyses – with the multilevel epigenetic data being the strongest part. The findings for adversity are not replicated and should therefore be interpreted more cautiously, with the reviewer's understanding that this is a uniquely phenotyped sample in this regard. Could the authors explore, whether at least some of the findings in the amygdala can also be seen in data from other brain regions from the same samples?

While this is the first study from our group that used WGBS and CHIP-Seq to investigate epigenetic consequences of early-life adversity (ELA), we nevertheless recently generated RNA-Seq data in another brain region, the anterior cingulate cortex (ACC, see¹). These data were obtained for a slightly larger cohort (n=50) that, importantly, included all individuals from the present amygdala study. Following the reviewer's suggestion, we compared gene expression changes across the 2 brain regions using the RRHO2 algorithm². Results uncovered strong patterns of common dysregulation, whereby large groups of genes show similar up- or downregulation in both regions in subjects with a history of ELA (see lower-left quadrant, UP/UP, and upper-right quadrant, DOWN/DOWN in new FigS17b, next page):

Using the 2 gene lists identified by RRHO2, corresponding to most significant overlaps among down- ($p\text{-adj}=10^{-487}$, Benjamini-Yekutieli) or upregulated ($p\text{-adj}=10^{-388}$) genes, we then conducted gene ontology (GO) enrichment (FigS17c). Strikingly, a large number of GO terms previously identified during multi-epigenetic analysis of the single amygdala dataset (see FigS17f) also emerged from the combined analysis of transcriptomes from both regions:

This included GO terms related to immune processes, small GTPase signaling, neuronal physiology (including terms related to the regulation of neuronal membrane potential), cellular adhesion, and the cytoskeleton (see also full results in new supplementary Table 16).

Overall, these results (now included in the revised manuscript; see last paragraph of the results section) provide evidence that, at least in the present cohort of individuals with a history of ELA, part of transcriptional changes observed in the amygdala also affect another brain region, the ACC, that significantly contributes to mood regulation and depression pathophysiology. Future studies will be necessary to better understand whether similar or divergent epigenetic processes underlie such effects across various brain regions as a function of ELA.

I have some minor comments.

Could the authors give a bit more information on the cases, i.e. type of abuse, specific depression diagnosis (recurrent, psychotic etc..) and means of suicide?

To characterize cases and controls, as well as histories of ELA, psychological autopsies and proxy-based interviews were conducted using notably the Childhood Experiences of Care and Abuse (CECA) questionnaire (which yields severity scores ranging from 1 to 4 for physical abuse or neglect, and from 1 to 6 for sexual abuse), complemented by information from medical charts and coroner records. This process, which is well established in the field to assess histories of ELA and has been validated by our group in the context of proxy-based

interviews^{3,4}, allowed us to identify 4 major types of adverse experiences: neglect, physical abuse, sexual abuse, and psychological abuse. Only cases with the maximum severity ratings of 1 or 2 for at least one type of abuse were included in the ELA group.

In our cohort, physical abuse was the most frequent type (present in 66.7% of subjects), followed by neglect (42.9%), sexual abuse (23.8%), and psychological abuse (19.0%). As expected for such severely affected individuals, almost half of them suffered from more than 1 type of adverse experience (42.9%). This information is now included in Supplementary Table 2 of the revised manuscript:

	Neglect (%)	Physical abuse (%)	Sexual abuse (%)	Psychological abuse (%)	>1 type of abuse (%)
C	0	0	0	0	0
ELA	42,9	66,7	23,8	19,0	42,9

All subjects from the ELA group were diagnosed with major depressive disorder at the time of death (using DSM-IV criteria and SCID-I interviews⁵, adapted for psychological autopsies), while subjects meeting criteria for depression with psychotic features were excluded. Means of suicide were the following (see Supplementary Table 2):

Mean	Number of subjects
Hanging	10
Drugs with sedative effects	8
Jumping	2
Shooting	1

Finally, subjects from the control group were individuals who died suddenly in work-related accidents, cardiovascular arrest, or in car accidents, with a negative history of ELA and no psychiatric diagnosis.

Even though pooling of samples for analyses is plausible due to the amount of material necessary for the different ChIP-Seq protocols, it is a limitation and should be discussed and directly mentioned in the results when mentioning the sample size and suppl. Tables 1 and 2. Also it should be more clearly stated that WGBS and RNA seq was performed in the individual samples, this difference is not very clear in the main text and should also be mentioned there. For RNA seq, the exact number of samples should be mentioned, and whether any sample was excluded for low RIN, this is not clear but relevant, especially once the data are compared to the pooled sequencing in ChIP-Seq.

The information on tissue pooling for ChIP-Seq experiments (previously in the methods section) now appears, as suggested by the Reviewer, at the beginning of the results section. We also clarify the fact that RNA-Seq and WGBS data were generated for each individual subject:

“Histone landscapes. Six histone modifications were assessed in depressed subjects with histories of ELA and healthy controls (C) with no such history (Supplementary Tables1-3). Because of the small size of the lateral amygdala, and the significant amount of tissue required for multiple immuno-precipitations and ChIP-seq analysis of 6 marks, tissues were distributed into 7 ELA and 4 C pools (see Supplementary Table2). In contrast, WGBS and RNA-Seq data (see below) were generated for each individual sample (C, n=17; ELA, n=21).”

No sample was excluded because of low RIN value, which is now indicated in the methods section.

The discussion should include a limitations section, mentioning the pooling approach, lack of single cell resolution and lack of replication at this point with a more cautious interpretation of the molecular and biological findings with regards to adversity.

As requested, we included a section on limitations in the revised discussion, as follows:

“Of note, this study has limitations. First, due to technical constraints at the beginning of the project, pools of amygdala tissue from several subjects were analyzed for ChIP-Seq, while RNA-Seq and WGBS were conducted separately for each sample. While this may have affected our results (for example, by obscuring subtle subject-specific histone changes not detected at pool level), the convergence of functional annotations observed across multiple types of data suggest a modest detrimental impact of the pooling approach. Second, the field of genomics is currently moving towards the molecular analysis of single-cells, with the hope of achieving higher resolution and better understanding of psychopathology. In comparison, the present work focused on bulk tissue only and, as such, may have missed epigenetic processes affecting individual or rare cell-types. Finally, all ELA subjects died during a major depressive episode by means of suicide. Therefore, it is possible that part of the molecular adaptations that we uncovered, and cautiously associate with ELA, contribute to these complex phenotypes rather than stem specifically from ELA. Exploring this hypothesis will require replication in larger cohorts and additional clinical groups (eg, depressed suicides with no history of ELA).”

Reviewer #2

This is a very innovative study on the lateral amygdala of subjects diagnosed with depression/suicide and a history of early life adversity and controls. While the overall sample size is small (N=21 and N=17 per group), the study is unique because: (i) it is focused on the lateral amygdala, a clinically extremely important brain region but, because of limitations in tissue quantities, barely explored in the field of psychiatric neurogenomics, (ii) 8 different epigenetic markings (6 histone modifications and CAC and CG methylation profiles) are explored plus RNA-seq and (iii) Douglas-Bell Canada brain bank is one of only few (maybe even the only one) brain bank worldwide to have built a collection of brains from subjects for early life adversity.

So, the study will have a significant impact in the field and will make an extremely important contribution.

We thank the reviewer for the very positive evaluation of our work.

My suggestions to the authors:

a) their findings, given that they examined 8 epigenetic markings, are as expected, 'multidimensional', but they should make in the discussion section a better effort to synthesize their findings, or at least provide as the last figure a schematic or illustrative overview summarizing all their main findings so that the Reader does not get overwhelmed with the many details of their findings.

We understand from the Reviewer's comment that our discussion needed improvements. To facilitate its reading, it now includes several references to specific figures and panels, and has also been significantly re-organized and modified to better synthesize main findings. This includes the following modifications:

- Similar to the results section, findings related to epigenetic interactions in the healthy brain are now described first, followed by the description of changes associated with

ELA (which implied reorganizing the first 4 paragraphs); this is now mentioned in the first and third paragraphs:

“Imaging studies have consistently demonstrated that ELA associates with impaired function of the amygdala. Here, going beyond previous studies, we conducted a comprehensive analysis of its potential molecular consequences in this brain region, across multiple transcriptional and epigenetic mechanisms. Below, we discuss implications of our results: first, in the healthy brain; second, in relation to ELA.”

“Beyond molecular interactions in physiological conditions, this study was primarily designed to investigate molecular consequences of ELA. Over the last two decades, (...)”

- As suggested, we included a new Figure (Fig8) to summarize differences between CG- and CAC forms of epigenetic plasticity associated with ELA:

b) technically: the Authors relied on Diffrep analysis tools to study histone case control differences. Which is fine, but keep in mind that Diffrep (if I am correctly informed) does not rely on peak calling algorithms but on sequence windows and may be prone to false positive findings, and thus their paper would benefit from additional analyses with peak calling, not to replace their Diffrep findings but to confirm and extend.

The Reviewer is correct as diffReps relies on a sliding window approach, as opposed to tools based on peak calling, such as MACS¹¹. While many approaches have been developed for differential analysis of ChIP-Seq data, little consensus has been achieved on what is the best strategy to use. Following recommendations from Steinhauser et al, who compared 14 different tools⁶, we opted for diffReps⁷, one of the most widely used⁸⁻¹⁰. To address the reviewer’s point, we have now applied a peak caller to our data, MACS, and identified peaks in each tissue pool for each histone mark. We then checked whether these peaks overlapped with the differential sites (DS) initially identified using diffReps when comparing C and ELA groups. This was the case for a large majority of all DS for 5 marks (from 71% to 88%, see Table below). The overlap was comparatively lower for H3K27me3, a mark that poorly contributed to differences observed across C and ELA group, either individually (see Fig.3b) or in terms of chromatin state transitions (see the Polycomb repressed state, Fig.4a). Using the R package regioneR¹², we then confirmed that overlap between diffReps-DS and MACS-peaks were strongly significant for all marks, including H3K27me3 (p<10E-05; 100,000

permutations). Overall, while we acknowledge that false positive are still possible at the level of individual genomic sites, these results indicate that our DS, collectively, are located in regions where histone modifications are abundant, and peaks are reliably identified.

Histone mark	Number of DS identified by diffReps	Number of DS intersecting peaks called using MACS2	Proportion of overlap	Z-score of overlap (regioneR, 10000 permutations)	Significance of overlap (regioneR, 10000 permutations)
H3K4me1	977	836	88.6%	58.8	p<10E-05
H3K4me3	584	418	71.4%	72.1	p<10E-05
H3K27ac	1524	1169	76.7%	79.4	p<10E-05
H3K36me3	951	681	71.6%	49.3	p<10E-05
H3K27me3	557	210	37.3%	85.2	p<10E-05
H3K9me3	533	475	88.2%	43.2	p<10E-05

c) there should be some discussion whether and how the pooling of samples from different individuals may or may not have affected the results. While such strategy may be unavoidable given the limited quantity of amygdala tissue, there should be a brief discussion in the Discussion section.

We agree with the Reviewer, and also with Reviewer 1, who expressed a similar concern. This limitation, among others, is now discussed in the revised discussion:

“Of note, this study has limitations. First, due to technical constraints at the beginning of the project, pools of amygdala tissue from several subjects were analyzed for ChIP-Seq, while RNA-Seq and WGBS were conducted separately for each sample. While this may have affected our results (for example, by obscuring subtle subject-specific histone changes not detected at pool level), the convergence of functional annotations observed across multiple types of data suggest a modest detrimental impact of the pooling approach. Second, the field of genomics is currently moving towards the molecular analysis of single-cells, with the hope of achieving higher resolution and better understanding of psychopathology. In comparison, the present work focused on bulk tissue only and, as such, may have missed epigenetic processes affecting individual or rare cell-types. Finally, all ELA subjects died during a major depressive episode by means of suicide. Therefore, it is possible that part of the molecular adaptations that we uncovered, and cautiously associate with ELA, contribute to these complex phenotypes rather than stem specifically from ELA. Exploring this hypothesis will require replication in larger cohorts and additional clinical groups (eg, depressed suicides with no history of ELA).”

References

1. Lutz, P. E. *et al.* Association of a History of Child Abuse With Impaired Myelination in the Anterior Cingulate Cortex: Convergent Epigenetic, Transcriptional, and Morphological Evidence. *Am J Psychiatry* 174, 1185–1194 (2017).
2. Plaisier, S. B., Taschereau, R., Wong, J. A. & Graeber, T. G. Rank-rank hypergeometric overlap: identification of statistically significant overlap between gene-expression signatures. *Nucleic Acids Res* 38, e169 (2010).
3. Tousignant, M. *et al.* Suicide, schizophrenia, and schizoid-type psychosis: role of life events and childhood factors. *Suicide Life Threat Behav* 41, 66–78 (2011).
4. Zouk, H., Tousignant, M., Seguin, M., Lesage, A. & Turecki, G. Characterization of impulsivity in suicide completers: clinical, behavioral and psychosocial dimensions. *J Affect Disord* 92, 195–204 (2006).
5. Spitzer, R. L., Williams, J. B., Gibbon, M. & First, M. B. The Structured Clinical Interview for DSM-III-R (SCID). I: History, rationale, and description. *Arch. Gen. Psychiatry*

49, 624–629 (1992).

6. Steinhauser, S., Kurzawa, N., Eils, R. & Herrmann, C. A comprehensive comparison of tools for differential ChIP-seq analysis. *Brief. Bioinformatics* 17, 953–966 (2016).
7. Shen, L. *et al.* diffReps: detecting differential chromatin modification sites from ChIP-seq data with biological replicates. *PLoS One* 8, e65598 (2013).
8. Farrelly, L. A. *et al.* Histone serotonylation is a permissive modification that enhances TFIID binding to H3K4me3. *Nature* 567, 535–539 (2019).
9. Berg, D. A. *et al.* A Common Embryonic Origin of Stem Cells Drives Developmental and Adult Neurogenesis. *Cell* 177, 654–668.e15 (2019).
10. Delgado, R. N. *et al.* Maintenance of neural stem cell positional identity by mixed-lineage leukemia 1. *Science* 368, 48–53 (2020).
11. Zhang, Y. *et al.* Model-based analysis of ChIP-Seq (MACS). *Genome Biol.* 9, R137 (2008).
12. Gel, B. *et al.* regioneR: an R/Bioconductor package for the association analysis of genomic regions based on permutation tests. *Bioinformatics* 32, 289–291 (2016).

Reviewer #1 (Remarks to the Author):

The authors have addressed all remaining concerns and the added data from the Acc strengthens the findings.

Reviewer #2 (Remarks to the Author):

The Authors have addressed adequately the issues raised in previous round of review. I have no concerns or additional issues.